



# Experiments with the modified Rotating Shallow Water model (modRSW, v.1.0): assessing the relevance for convective-scale data assimilation research

Thomas Kent[1,2,†,*], Luca Cantarello[1,2,♣,*], Gordon Inverarity[3], Steven Tobias[1,2], and Onno Bokhove[1,2]

[1]School of Mathematics, University of Leeds, UK
[2]Leeds Institute for Fluid Dynamics, University of Leeds, UK
[3]Met Office, Exeter, UK
[†]Currently at: Norlys Energy Trading A/S, Aalborg, Denmark
[♣]Currently at: European Centre for Medium-Range Weather Forecasts (ECMWF), Bonn, Germany
[*]These authors contributed equally to this work.

**Correspondence:** luca.cantarello@ecmwf.int & o.bokhove@leeds.ac.uk

**Abstract.** Following the development of the modified rotating shallow water model (modRSW), which includes simplified dynamics of convection and precipitation, we present the results of a series of idealised forecast-assimilation experiments to demonstrate its relevance for data assimilation research at convective scales, focusing on its ability to imitate an operational Numerical Weather Prediction system. We address in a rigorous manner how to ascertain whether an idealised model is relevant

for data assimilation research by comparing our modRSW model configuration – based on a twin-setting approach in which synthetic observations are assimilated hourly into a Deterministic Ensemble Kalman filter – with the most common properties of an operational system. We demonstrate forecast-assimilation experiments that produce values of error growth rates (6-9 hours) and observational influence on the analyses (around $30\%$) comparable to those found in operational systems. In addition, we provide a description of the approach we took to reach a well-tuned configuration by comparing the ensemble spread with

the Root Mean Square Error of the ensemble mean, and examining the Continuous Ranked Probability Score. We also provide subjective assessments of forecasts at different lead times.

## 1 Introduction

Idealised models of the atmosphere are often used to conduct inexpensive data assimilation (DA) research in meteorology and oceanography. They represent a convenient alternative to computationally expensive operational Numerical Weather Prediction

(NWP) models.

Several idealised and simplified models are already available. Some of these are elementary models describing only basic processes related to the atmosphere, such as those famously developed by Lorenz (Lorenz, 1963, 1996, 2005) to investigate the dynamics of deterministic non–periodic (in time) systems and their predictability. Other models are more complex and represent a compromise between reducing the computational cost of an NWP model and retaining a closer connection with the

equations governing the atmosphere; some of these models have been explicitly developed for data assimilation research and





include the non-hydrostatic 'ABC' model for convective-scale DA research (Petrie et al., 2017; Bannister, 2020), the model of intermediate complexity developed by Ehrendorfer and Errico (2008) and the four-dimensional Moist Atmosphere Dynamics Data Assimilation Model (MADDAM, see Zaplotnik et al. (2018)).

One class of particularly well-studied idealised models is represented by shallow water models (Zeitlin, 2018). Such mod-
els have been used extensively in data assimilation research for both meteorological and oceanographic applications (Žagar et al., 2004; Salman et al., 2006; Stewart et al., 2013). In this paper, we present a series of forecast-assimilation experiments conducted with the modified Rotating Shallow Water model (modRSW) developed by Kent et al. (2017) for convective-scale DA research. The modRSW model is an improved (hyperbolic and rotating) version of the simpler cumulus convection model developed by Würsch and Craig (2014) – hereafter W&C – which has already been used extensively in many DA studies
(Ruckstuhl and Janjić, 2018; Zeng et al., 2018, 2019). A revised, isentropic and multi-layer version of the modRSW model (is-modRSW) has also been developed (Cantarello et al., 2022; Bokhove et al., 2022) and used in Cantarello (2021) to investigate idealised satellite DA.

The key feature that makes an idealised model valuable for operational-oriented research is its *relevance*, in particular its ability to retain some of the fundamental characteristics and behaviours of operational NWP systems, despite the overall reduction in
complexity caused by the use of a simpler configuration. Unfortunately, assessing the relevance of idealised models for DA applications is something that is often overlooked in the literature, despite a large and growing body of research in DA.

The purpose of this paper is therefore to assess the performance and the relevance of the modRSW model for convective-scale data assimilation research with the aid of a series of well-tuned forecast-assimilation experiments based on a Deterministic Ensemble Kalman Filter (DEnKF, Sakov and Oke (2008)). The characteristics of the modRSW model and the results of the
experiments are compared with those found in most operational NWP systems, highlighting the value of this model as a suit-able tool to conduct convective-scale DA research. In addition, we have developed an objective and detailed *tuning* procedure with the aim of finding the best filter configuration for a simplified observing system, mimicking the work done routinely in any operational weather service. This procedure could also be applied to other idealised models.

In an operational forecast–assimilation system, tuning is performed to produce the lowest forecast error over a range of lead
times for fields of interest given the available observing system. Typically, this involves permuting through various parameters associated with assimilation algorithms, such as ensemble size, inflation factors/methods and localisation length-scales, in an attempt to arrive at the filter configuration with the best performance. The computational cost and time taken to run NWP ex-periments with operational systems means that it is not feasible to test every possible combination of parameters. Historically, choices have been guided by intuition and experience, while using a full set of objective measures to evaluate each config-
uration (e.g., Bowler et al. (2017)). Recent attempts at optimal tuning by Ménétrier et al. (2015a,b) pursue a more objective approach than simply permuting through a prescribed set of parameters. We note also that tuning in an operational setting often comprises a combination of objective and subjective verification.

The process of tuning an idealized forecast–assimilation system differs somewhat from the operational case in that we have the freedom to choose the observing system. How it is generated should reflect the problem at hand and in some sense becomes
part of the process: the observing system should be tuned alongside the filter configuration to produce an idealized system





that demonstrates attributes of an operational system. For example, if an experiment is deemed optimal in a minimum-error sense but has an observational influence of, say, a few percent, it cannot be considered relevant since convective-scale NWP is expected to have average observational influence $\leq 40\%$ (section 4.3.3). Likewise, if error growth rates of ensemble forecasts initialised using optimal analysis ensembles are not comparable with operational values, it is difficult to consider the experi-

ment meaningful from an NWP perspective.

In this paper, we present a systematic tuning process by following a series of steps that helps narrow down the tuning parameter space until a specific set of filter parameters is selected. Finally, a single, well-tuned experiment is subsequently analysed to further demonstrate its relevance for convective-scale DA.

The structure of the paper is as follows: in section 2, we present a comparison between the characteristics of the idealised setup

used in this paper and those of a typical operational NWP system. In section 3, we describe briefly the modRSW model and the twin-setting configuration used to conduct DA experiments and the Deterministic Ensemble Kalman filter (DEnKF) that is used to update the ensemble of forecasts every hour. In section 4, we discuss the experimental setup and diagnostics used to evaluate the results. In section 5, we report the results of the forecast-assimilation experiments, discussing our tuning strategy to achieve an optimal configuration and showing that the modRSW model is relevant for convective-scale data assimilation

research. We conclude in section 6 with a holistic summary of our study and highlight its novelty in the field of idealised DA research.

To facilitate the dissemination of the model and its DA system, this study is complemented with the Python code used for running the experiments, evaluating performance and relevance, and plotting the results. The code is published online in an open-source GitHub repository, and forms part of a suite of idealised DA projects developed by the authors; more details are

given in the Code Availability section.

## 2 Comparing idealised and operational systems

At the core of this paper is the idea that it is important for an idealised model for DA research to be *relevant* to an operational forecast-assimilation system. In this regard, by combining the strong nonlinearity due to the onset of convection and precip-

itation and the hydrodynamic (advective) nonlinearity of the shallow water equations, the modRSW model captures some fundamental dynamical processes of convective-scale weather systems and provides an interesting and useful testbed for data assimilation research at convective scales (Kent, 2016; Kent et al., 2017). This has also been recognised in studies that, for example, implement the model directly (Kriegmair et al., 2022), develop a variation of the model for satellite DA research (Bokhove et al., 2022; Cantarello et al., 2022), and propose its use in other idealised DA studies (Fowler, 2019).

We will substantiate this further in this article by describing in detail the process of arriving at a well-tuned and relevant set of idealised forecast–assimilation experiments and thoroughly assessing their performance via objective metrics and subjective verification. However, we want to define in advance the approach followed in constructing our idealised experiments, in a way that helps address the model's relevance for convective–scale DA and NWP. The essence of this approach is based on



**Table 1.** Idealised forecast–assimilation experiments for convective–scale Numerical Weather Prediction: protocol, results, and relevance. The protocol is summarised in the first two columns: the first column lists aspects of forecast–assimilation systems that idealised experiments should seek to replicate where possible; the second column gives typical values for an operational system; column 3 gives the values achieved for a well-tuned idealised experiment using the modRSW model and the ensemble Kalman filter, detailed in section 5; the last column appraises the relevance of these results by comparing the values in columns 2 and 3 and attributing medium (–) or high (✓) relevance where applicable, with N/A used for quantities that are deliberately chosen to be different in order to attain idealised status.

| | Operational system | Our idealised system | Relevant? |
|---|---|---|---|
| *Prescribed parameters* | | | |
| Forecast resolution | $\mathcal{O}(1\,\text{km})$ | 2.5 km | ✓ |
| Update frequency | $\mathcal{O}(1\,\text{hr})$ | 1 hr | ✓ |
| Ensemble size, $N$ | $\mathcal{O}(10-100)$ | 18 | ✓ |
| Number of observations, $p$ | $\mathcal{O}(10^7)$ | 28 | N/A |
| State dimension, $n$ | $\mathcal{O}(10^9)$ | 600 | N/A |
| Rank-deficiency | $N \ll p \ll n$ | $N < p < n$ | ✓ |
| Observation operator | Nonlinear | Linear | – |
| *Tuning parameters* | | | |
| Observing system: | | | |
| (i) observation density | Varied | $\sim 50$ km | ✓ |
| (ii) observation error | Correlated and uncorrelated | Uncorrelated | – |
| Filter configuration: | | | |
| (i) localisation (horiz.) | Lengthscale: $\mathcal{O}(100\,\text{km})$ (obs. space) | $\sim 500$ km (model-space) | ✓ |
| (ii) Inflation | Adaptive (RTPP; RTPS) and additive[1] | $\alpha_{\text{RTPP}} = 0.5, \alpha_{\text{RTPS}} = 0.7, \gamma_a = 0.15$ | ✓ |
| *Tuning criteria* | | | |
| (i) RMSE | $\sim 1$ | $\sim 1$ | ✓ |
| (ii) RSME | Minimum for given lead times | Minimum for 3 hr forecast | – |
| (iii) CRPS | Minimum for given lead times | Minimum for 3 hr forecast | – |
| *Validation criteria* | | | |
| Observational influence | Global: $\sim 20\%$; high-res.: $> 20\%$ | $\sim 30\%$ | ✓ |
| Error-doubling time $T_d$ | Global: $\mathcal{O}(1\,\text{day})$; high-res.: $\mathcal{O}(1\,\text{hr})$ | $\sim 6-9$ hrs | ✓ |

[1] Owing to the many different approaches in operational systems, it is not possible to give precise target values for inflation factors. In the literature, we find values for Relaxation to Prior Perturbations in the region of $\alpha_{\text{RTPP}} = 0.5 - 0.75$ and for Relaxation to Prior Spread $\alpha_{\text{RTPS}} = 0.7 - 0.95$ (e.g., Bick et al. (2016); Schraff et al. (2016); Gustafsson et al. (2018); Inverarity et al. (2022)), for both global and convective-scale systems.

a side-by-side comparison between operational systems and our idealised model, which is summarised in Table 1. The table

lists aspects of forecast–assimilation systems that idealised systems should seek to replicate where possible, along with criteria



to guide the tuning and validation process; each aspect is then supplemented by its typical values for an operational system (where applicable, cf. Gustafsson et al. (2018)) and its value in our idealised model for comparison.

For this demonstration, we have chosen a model resolution and cycling frequency (Table 1) that are directly comparable with operational systems (see, e.g., Table 5 in Gustafsson et al. (2018)). An idealised system has, by construction, fewer degrees of freedom ($n$) and number of observations ($p$) than than an operational system; however, operational systems are characterised by their rank-deficiency, and this is imposed on our idealised system by choosing the ensemble size $N$ to satisfy $N < p < n$. The observing system is chosen here to be a simplification (linear operator, uncorrelated errors) whilst having enough flexibility (through observation density and error magnitude) to tune the system, noting that the realistic treatment of observations is not the focus of this study. The choice of $50\,\mathrm{km}$ as the average spacing between observations is, however, in line with the density of UK land observations[1] of around $40\,\mathrm{km}$.

Operational data assimilation techniques that use ensemble information to describe forecast-error covariances typically employ various techniques to account for different sources of error (Houtekamer and Zhang, 2016). Forecast model error is accounted for with additive inflation in the forecast model. Sampling error resulting from the limited number of ensemble members leads to spurious noise in the estimated covariances, which is then reduced by applying localisation to limit the spatial influence of an observation. Finally, another impact of a having a small ensemble is that the ensemble is typically underspread following the data assimilation step, so relaxation techniques (such as Relaxation to Prior Perturbations, RTPP, and Relaxation to Prior Spread, RTPS) are used to reinflate the ensemble. Each of these techniques has an associated parameter that needs to be tuned, and the combination of these parameters here defines the filter configuration. These are then compared to typical operational values in Table 1 to show the relevance of the resulting idealised configuration.

For an experiment to be considered well-tuned, the ensemble spread (SPR; i.e., the root mean square difference between the ensemble members and the ensemble mean) should be comparable to the root mean square error (RMSE) of the ensemble mean (cf. Whitaker and Loughe (1998)). Thus, we demand the ratio $\mathrm{SPR}/\mathrm{RMSE} \sim 1$ and seek to minimise the RMSE and the Continuous Ranked Probability Score (CRPS; Hersbach (2000)), which measures the the accuracy of the full (discrete) distribution implied by ensemble. Experiments are optimised on forecasts with a 3-hour lead-time (denoted as T+3) since we do not necessarily want to produce the best analysis but the best forecast (lowest RMSE) at a desired lead time, noting that 3-hour forecasts are within the 6 hours that conventionally define the nowcasting range (Ballard et al., 2012; Sun et al., 2014).

If an idealised system achieves all of the above, it is here validated for its relevance for convective-scale NWP primarily via the overall Observation Influence Diagnostic (OID) measure and error-doubling time statistics, following earlier work by Inverarity (2015) on deterministic DA using an idealised model. Alongside objective verification measures for system performance, subjective assessment of model output is common in operational NWP. Therefore, we also examine a well-tuned and relevant case via a subjective visual assessment, giving a more holistic illustration of the impact of cycled data assimilation on forecast model fields.

---

[1]https://www.metoffice.gov.uk/weather/guides/observations/uk-observations-network.



## 3 Data assimilation experiments with the modRSW model

In this section, we describe the general setup used to perform idealised data assimilation experiments using the modRSW
model and a Deterministic Ensemble Kalman filter (DEnKF). To facilitate the understanding of the rest of the manuscript, we
introduce and summarise here the notation utilised in this paper, which follows where possible the notation of Houtekamer and
Zhang (2016).

The $n$-dimensional state vector, consisting of a column of discretised model variables, is denoted by $\mathbf{x} \in \mathbb{R}^n$ and the $p$-
dimensional vector of observations is denoted $\mathbf{y} \in \mathbb{R}^p$. Superscripts 'f' and 'a' applied to the state vector indicate the forecast
and analysis state, respectively, whereas an overbar $\overline{(\cdot)}$ indicates an ensemble mean. Superscript 'T' denotes matrix trans-
position or conversion between a row and column vector. A linear observation operator[2] $\mathbf{H} : \mathbb{R}^n \to \mathbb{R}^p$ generates simulated
observations by mapping the state vector $\mathbf{x}$ from model to observation space; the nonlinear model operator $\mathcal{M} : \mathbb{R}^n \to \mathbb{R}^n$ is
the numerical discretisation of the modRSW model.

### 3.1 The modRSW model

Our experiments are performed with the modRSW model, which is described in detail in Kent (2016) and Kent et al. (2017),
including finite element numerical implementation details.

The modRSW model introduces a number of alterations to the classic shallow water equations in regions where the fluid
exceeds two threshold heights. These thresholds provide highly nonlinear switches for the onset of convection and precipitation,
enabling a simplified representation of cumulus convection to be incorporated in the model without explicitly considering
temperature and other thermodynamic properties. An extra variable is also introduced for the idealised transport of moisture
(the 'rain mass fraction' $r$), which is coupled to the horizontal momentum equation. The modRSW model is described by the
following equations:

$$\partial_t h + \partial_x(hu) = 0, \tag{1a}$$

$$\partial_t(hu) + \partial_x(hu^2 + P) + hc_0^2\partial_x r - fhv = -Q\partial_x b, \tag{1b}$$

$$\partial_t(hv) + \partial_x(huv) + fhu = 0, \tag{1c}$$

$$\partial_t(hr) + \partial_x(hur) + h\widetilde{\beta}\partial_x u + \alpha hr = 0, \tag{1d}$$

where $h = h(x,t)$ is the fluid depth, $b = b(x)$ is the prescribed underlying topography (so that $h + b$ is the free-surface height),
$u(x,t)$ and $v(x,t)$ are velocity components in the zonal $x$– and meridional $y$–directions, $f$ is the Coriolis parameter (typically
$10^{-4}\mathrm{s}^{-1}$ for midlatitudes) and $t$ is time. $P$ and $Q$ are defined via the effective pressure $p = p(h) = \frac{1}{2}gh^2$, where $g = 9.81\mathrm{ms}^{-2}$

---

[2]The EnKF equations can be derived for a fully nonlinear observation operator $\mathcal{H}$ but, since a linear operator is used in this study, we continue with $\mathbf{H}$
only.



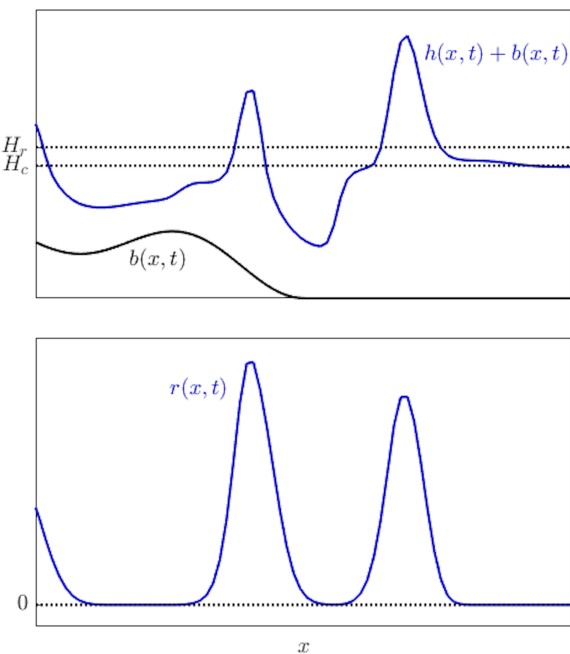

**Figure 1.** Illustrative solutions of the free surface height $h+b$ (top) and model 'rain' $r$ (bottom). The black dotted lines in the top panel denote the threshold heights $H_c < H_r$, above which the classical shallow water dynamics are modified by the switches in Eqs. 2 and 3, leading to a simplified representation of convection and associated precipitation. Note that b is independent of time in the experiments considered in this paper.

is the gravitational acceleration, by:

$$P(h;b) = \begin{cases} p(H_c - b), & \text{for } h + b > H_c, \\ p(h), & \text{otherwise,} \end{cases} \tag{2a}$$

$$Q(h;b) = \begin{cases} p'(H_c - b), & \text{for } h + b > H_c, \\ p'(h), & \text{otherwise,} \end{cases} \tag{2b}$$

with $p'$ denoting the derivative of $p$ with respect to its argument $h$, and the 'rain switch':

$$\widetilde{\beta} = \begin{cases} \beta, & \text{for } h + b > H_r \text{ and } \partial_x u < 0, \\ 0, & \text{otherwise.} \end{cases} \tag{3}$$

The threshold heights $H_c$ and $H_r > H_c$ (units m), appearing in the modified pressure terms (Eq. 2) and 'rain' switch term (Eq. 3), correspond to the onset of convection and precipitation, respectively. The constants $\alpha > 0$ (units $\mathrm{s}^{-1}$) and $\beta > 0$ (dimensionless) control the removal and production of model 'rain' respectively; $c_0^2$ (units $\mathrm{m}^2\mathrm{s}^{-2}$) converts the dimensionless





$r$ into a potential in the momentum equation and controls the strength of the feedback (cf. Würsch and Craig (2014)). Schematic solutions for $h$ and $r$ and the threshold heights are illustrated in Fig. 1.

Würsch and Craig (2014) introduced the model 'rain', or 'rain mass fraction' $r$, as an artificial representation of precipitation: it should be regarded as the fraction of mass in a column that has precipitated. Kent et al. (2017) further emphasised that it is never removed from the system (total mass is conserved according to Eq. 1a), with the sink term $\alpha hr$ in Eq. 1d transferring mass from a precipitated state $r$ to a notional precipitable state $1 - r$ so that it may precipitate again at a later time. This is important for idealised DA experiments as it means that the model does not run out of moisture and so can continues to

precipitate for long forecast lead times.

It is sometimes convenient to work with the non-dimensional form of the system of equations in Eq. (1):

$$\partial_t h + \partial_x (hu) = 0, \tag{4a}$$

$$\partial_t (hu) + \partial_x (hu^2 + P) + h\widetilde{c_0}^2 \partial_x r - \frac{1}{\mathrm{Ro}} hv = -Q\partial_x b, \tag{4b}$$

$$\partial_t (hv) + \partial_x (huv) + \frac{1}{\mathrm{Ro}} hu = 0, \tag{4c}$$

$$\partial_t (hr) + \partial_x (hur) + h\widetilde{\beta} \partial_x u + \widetilde{\alpha} hr = 0, \tag{4d}$$

in which we have scaled $x$ by the domain width $L_0$, $h$, $b$, $H_c$ and $H_r$ by a characteristic depth $H_0$; $u$ and $v$ by a characteristic horizontal speed $V_0$; time by $L_0/V_0$; and, the pressure appearing in Eq. (2) by $gH_0^2$ (Table 2). This in turn leads to the following dimensionless parameters, including the Rossby number Ro and Froude number Fr, with the latter appearing in the dimensionless P and Q functions, which are obtained by scaling the pressure and its gradient in Eq. (2) and then dividing both

expressions by $\mathrm{Fr}^2$:

$$\mathrm{Fr} = \frac{V_0}{\sqrt{gH_0}}, \quad \mathrm{Ro} = \frac{V_0}{fL_0}, \quad \widetilde{c_0}^2 = \frac{c_0^2}{V_0^2}, \quad \widetilde{\alpha} = \frac{L_0}{V_0}\alpha. \tag{5}$$

We refer the reader to Appendix 1 of Kent et al. (2017) for a detailed derivation of the non-dimensional Eq. (4).

### 3.2   Twin-setting configuration: nature run, observations and forecasts

DA schemes for NWP models are based on two main sources of information: a-priori knowledge of the atmospheric conditions,

or background state (typically coming from an earlier model forecast) and a set of recent observations. In the case of the modRSW model (and similarly to other idealised models), the so-called *twin-setting* configuration is used, by which the same computational model is used to generate both the nature run (which acts as a surrogate truth) and the forecasts.

The nature run is a single, long integration of the numerical model and is deemed to represent the true evolution in time of the physical system. Unlike in operational NWP, it provides a verifying data set with which to compare the forecast and

analaysis estimates and thus quantify the errors in each. Furthermore, it is used to produce *synthetic observations* of the physical system, which are then assimilated into the cycled forecast-assimilation system. These synthetic observations are generated by applying the observation operator $\mathbf{H}$ to the state vector from the nature run $\mathbf{x}^{\mathrm{nat}}$ (in our simple setup, this equates to sub-sampling the nature run at specific locations along the domain) and adding random noise drawn from a Gaussian observational





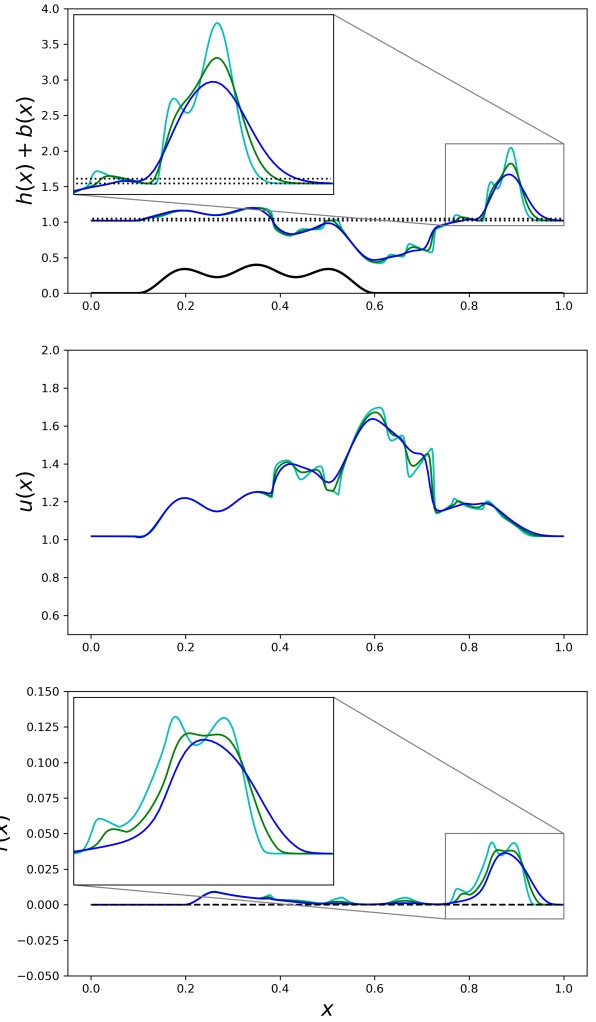

**Figure 2.** Model solutions for $h$ (top), $u$ (middle), and $r$ (bottom), with zoomed insets for $h$ and $r$, for $n_x = 200$ (blue), 400 (green) and 800 (cyan) grid points. In an imperfect model scenario, the forecast model has a coarser resolution than the nature run. Top panel: the thick black line is the topography $b(x)$ (Eq. 14) and the black horizontal dotted lines are the threshold heights $H_c < H_r$. Bottom panel: the black dashed line is $r(x) = 0$, confirming that the numerical integration scheme ensures non-negativity of $r$.

error distribution.

The twin-setting configuration used in this paper is based on a so-called 'imperfect' model scenario, in which the nature run is generated at twice the resolution of the forecast model, resulting in $n_x^{\mathrm{nat}} = 400$ evenly spaced horizontal grid points in the nature run and $n_x = 200$ for the forecasts, spanning the same 500 km domain. This choice reflects the fact that, despite improved representation of clouds and precipitation in NWP models with grid spacings of $\mathcal{O}(1 \text{ km})$, it is widely recognised that





convection is still under-resolved and does not exhibit all aspects of observed convection (Tang et al., 2013). Example model
trajectories at different resolutions at a given time are shown in Fig. 2. Conceptually, the basic data assimilation problem can
be summarised using this figure: adjust the forecast (low-resolution field) using synthetic observations based on the nature run
(high-resolution field) in order to provide a better estimate of the nature run using the imperfect forecast model.

### 3.3 The analysis step: implementing a Deterministic Ensemble Kalman Filter (DEnKF)

The DA algorithm used in this paper is the Deterministic Ensemble Kalman Filter (DEnKF) developed by Sakov and Oke
(2008). Along with other square root algorithms, the DEnKF avoids introducing additional sampling error associated with
perturbed observations (section 4.2). Furthermore, it is straightforward to implement and shares the EnKF's versatility and
effectiveness.

The DEnKF scheme calculates the analysis mean $\overline{\mathbf{x}}^{\mathrm{a}}$ via the standard Kalman update step:

$$\overline{\mathbf{x}}^{\mathrm{a}} = \overline{\mathbf{x}}^{\mathrm{f}} + \mathbf{K}_e(\mathbf{y} - \mathbf{H}\overline{\mathbf{x}}^{\mathrm{f}}), \tag{6a}$$

$$\mathbf{K}_e = \mathbf{P}_e^{\mathrm{f}}\mathbf{H}^T(\mathbf{H}\mathbf{P}_e^{\mathrm{f}}\mathbf{H}^T + \mathbf{R})^{-1}, \tag{6b}$$

with (ensemble) forecast-error covariance matrix $\mathbf{P}_e^{\mathrm{f}}$:

$$\mathbf{P}_e^{\mathrm{f}} = \frac{1}{N-1}\sum_{j=1}^{N}(\mathbf{x}_j^{\mathrm{f}} - \overline{\mathbf{x}}^{\mathrm{f}})(\mathbf{x}_j^{\mathrm{f}} - \overline{\mathbf{x}}^{\mathrm{f}})^T = \frac{1}{N-1}\mathbf{X}^{\mathrm{f}}(\mathbf{X}^{\mathrm{f}})^T, \tag{7}$$

where $\mathbf{X}^{\mathrm{f}}$ is the matrix whose $j^{\mathrm{th}}$ column consists of the ensemble perturbation $(\mathbf{x}_j^{\mathrm{f}})' = \mathbf{x}_j^{\mathrm{f}} - \overline{\mathbf{x}}^{\mathrm{f}}$, where subscript $j$ denotes the
$j^{\mathrm{th}}$ ensemble member. The analysis perturbations are updated as follows:

$$\mathbf{X}^{\mathrm{a}} = \mathbf{X}^{\mathrm{f}} - \frac{1}{2}\mathbf{K}_e\mathbf{H}\mathbf{X}^{\mathrm{f}}. \tag{8}$$

Each analysis state is then augmented with the mean to complete the process:

$$\mathbf{x}_j^{\mathrm{a}} = (\mathbf{X}^{\mathrm{a}})_j + \overline{\mathbf{x}}^{\mathrm{a}}, \tag{9}$$

where $(\mathbf{X}^{\mathrm{a}})_j$ is the $j^{\mathrm{th}}$ column of the matrix $\mathbf{X}^{\mathrm{a}}$.

Our implementation of the DEnKF departs slightly from the algorithm outlined in Sakov and Oke (2008), although it is based
on an equivalent generalised approach which is described in detail in Appendix A.

Additional techniques (i.e. localisation, ensemble inflation and self-exclusion) are required to compensate for the sampling
error associated with the finite ensemble size; in severely rank-deficient cases (i.e., when $N \ll n$), they are crucial for main-
taining satisfactory filter performance, particularly for nonlinear systems. The downside of using relatively *ad hoc* inflation
and localisation techniques is that there are several parameters that need tuning; the filter configuration in our idealised frame-
work is defined by the combination of parameters $L_{\mathrm{loc}}$, $\gamma_a$, and $\alpha_{\mathrm{RTPS}}$ for localisation, additive inflation, and RTPS inflation
respectively.

Here we address each in turn and describe the approach taken in our experiments.

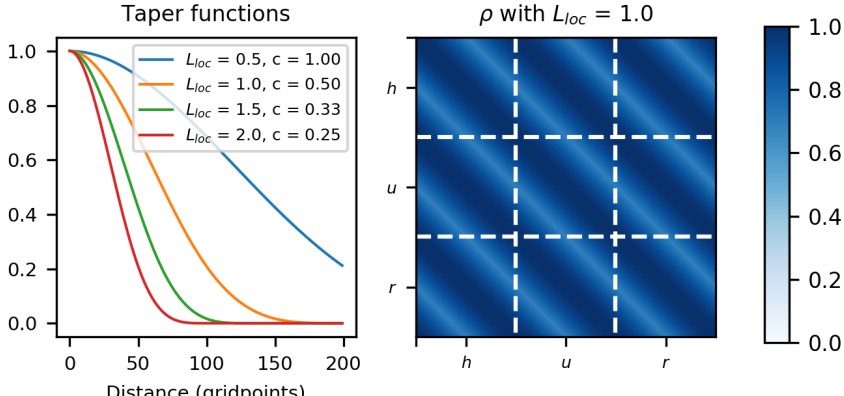

**Figure 3.** Left: Gaspari-Cohn taper function $\rho$, showing correlation on the vertical axis as a function of distance in model-space with $n_x = 200$ grid points (horizontal axis) for various $L_{\mathrm{loc}}$ and equivalent $c$ values. The localisation parameter $L_{\mathrm{loc}} = 1$ (orange) defines a cut-off length of $n_x / L_{\mathrm{loc}} = 200$ grid points (associated with $c = 0.5$), beyond which correlations are zero. Increasing $L_{\mathrm{loc}}$ (green and red curves) leads to a more severe localisation. Right: banded localisation matrix $\boldsymbol{\rho}$ with $L_{\mathrm{loc}} = 1.0$, which has the same dimension as $\mathbf{P}_e^f$ and is a block $3 \times 3$ matrix for our three-variable system.

### 3.3.1 Localisation

Localisation attempts to prevent the analysis estimate being degraded by suppressing spurious long-range correlations in the

forecast-error covariance matrix (Hamill et al., 2001; Houtekamer and Mitchell, 2001; Whitaker and Hamill, 2002). In model-space covariance localisation (cf. Carrassi et al. (2018)), this is achieved by multiplying the elements of the forecast-error covariance matrix with elements of a covariance taper matrix $\boldsymbol{\rho} \in \mathbb{R}^{n \times n}$ that reduces correlations as a function of distance. Entries of the covariance taper matrix $\boldsymbol{\rho}$ are calculated using a correlation function $\rho$ with compact support (i.e., non-zero in a local region, zero everywhere else), resulting in a localised forecast-error covariance matrix $\mathbf{P}_{\mathrm{loc}}^f = \boldsymbol{\rho} \circ \mathbf{P}_e^f$, where $\circ$ is the

Schur product[3].

Figure 3 illustrates the process of covariance localisation[4] in our system with 200 grid points. The correlations are filtered out gradually and suppressed completely (due to the compact support) beyond a certain distance, after which an observation has no influence. The localisation function plotted in Fig. 3 is the one proposed by Gaspari and Cohn (1999), which is reported for convenience in appendix C. In keeping with the dimensionless equations used in this paper, we define parameter $L_{\mathrm{loc}} = $

$1/(2c)$, where $c$ is the dimensionless Gaspari-Cohn distance parameter (Appendix C). Larger values of $L_{\mathrm{loc}}$ indicate that more localisation is being applied, with $1/L_{\mathrm{loc}}$ corresponding to the fractional domain size after which correlations are reduced to

---

[3]In matrix operations, this comprises elementwise multiplication $(\mathbf{A} \circ \mathbf{B})_{ij} = \mathbf{A}_{ij}\mathbf{B}_{ij}$ for two matrices $\mathbf{A}$ and $\mathbf{B}$ of the same dimension and $i, j$ indexing the row and column number respectively (Schur, 1911).

[4]An attractive feature of the DEnKF is that it readily permits a Schur-product-based (i.e., model space) covariance localisation.



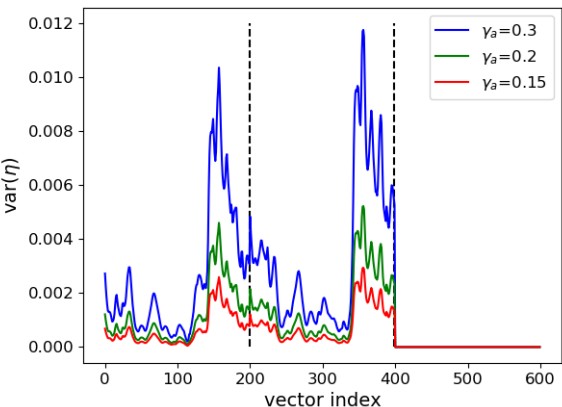

**Figure 4.** Spatial structure of the (variance of the) model-error perturbation vector $\boldsymbol{\eta}_j \sim \mathcal{N}(\mathbf{0}, \gamma_a^2 \mathbf{Q})$ used in additive inflation (Eq. 10). The plot shows $\mathrm{var}(\boldsymbol{\eta}_j) = \gamma_a^2 \mathrm{diag}(\mathbf{Q})$, a vector of dimension $n = 600$, for three candidate values of the scaling factor $\gamma_a$ used in the tuning process. The vertical dashed lines partition the vector into three for the $h$-, $hu$-, and $hr$-components of the variance, respectively. The model-error covariance matrix $\mathbf{Q}$ is diagonal.

zero. Thus, $L_{\mathrm{loc}} = 1$ corresponds to the domain size, while $L_{\mathrm{loc}} = 2$ means that correlations will be suppressed after half the domain size.

### 3.3.2 Ensemble inflation

Ensemble inflation techniques attempt to maintain a sufficient ensemble spread (and satisfactory filter performance) by artificially increasing the spread of ensemble members; typically, a combination of additive, multiplicative, and/or adaptive methods is used in practice. However, it should be noted that the alterations in the ensemble trajectories due to inflation dilute the impact of flow–dependent statistics developed in the EnKF (cf. Houtekamer and Zhang (2016)). We employ a combination of adaptive and additive inflation, as described below

**Additive inflation**

Additive inflation comprises adding random Gaussian perturbations $\boldsymbol{\eta}_j \sim \mathcal{N}(0, \gamma_a^2 \mathbf{Q})$ during the forecast step:

$$\mathbf{x}_j^{\mathrm{f}}(t_i) = \mathcal{M}(\mathbf{x}_j^{\mathrm{a}}(t_{i-1})) + \boldsymbol{\eta}_j, \quad j = 1, ..., N, \tag{10}$$

where the model-error covariance matrix $\mathbf{Q}$ is prescribed from some knowledge of the modelling system and $\gamma_a$ is a tuneable parameter controlling the overall magnitude of the sample perturbations. The $\gamma_a$ parameter is used to adjust the amount of

sampled additive noise in recognition of the fact that estimates for $\mathbf{Q}$ are inherently approximate, and mainly intended to represent the relative magnitudes of and correlations between the different components. How one best defines $\mathbf{Q}$ is an open question - ideally it should be constructed using flow-dependent perturbations (Hamill and Whitaker, 2011) but it is often a





static matrix precomputed from historical data. Since the aim of this study is not to investigate the role of the $\mathbf{Q}$ matrix, we adopt the most straightforward approach for an idealised setup, generating the model-error covariance matrix by exploiting

the difference in resolution between the nature run and the forecast model. In particular, we define the model error $\boldsymbol{\eta}_j$ as the difference between a deterministic model run and the nature run trajectory at each grid point for a one-hour forecast (matching the assimilation frequency), with both forecasts starting from the same initial conditions taken from the nature run. From the error distribution obtained with 48 forecast pairs, each starting with a different initial condition at different times, we calibrate our model-error covariance matrix $\mathbf{Q}$ using the unbiased sample covariance estimator with a denominator of $N-1$

and neglecting all the non-diagonal terms for simplicity. We also set to zero the diagonal terms relating to the model rain $r$, as it is nonlinearly related to $h$ and inflating both might cause $r$ to be over-inflated. To avoid introducing bias into the model state $\mathbf{x}(t_i)$, we draw a random sample $\tilde{\boldsymbol{\eta}}_j$ for $j=1,..,N$ and then adjust each member so that the sample has zero mean with:

$$\boldsymbol{\eta}_j = \tilde{\boldsymbol{\eta}}_j - \frac{1}{N}\sum_{i=1}^{N}\tilde{\boldsymbol{\eta}}_i. \qquad (11)$$

A graphical representation of the diagonal model-error covariance matrix $\mathbf{Q}$ is shown in Fig. 4. We note the geographical

variation in the model-error variance, with larger errors downstream of the topography (where much of the convection is triggered, cf. top panel in Fig. 2), and emphasise the approximate nature of this derived $\mathbf{Q}$ for the purpose of additive inflation tuning. Motivated by Bowler et al. (2017), additive inflation is implemented in Eq. (10) via the Incremental Analysis Update method (IAU, see Bloom et al. (1996)) by spreading $\boldsymbol{\eta}_j$ contributions appropriately (i.e., proportional to the model's adaptive time-step) throughout the numerical integration from $t_{i-1}$ to $t_i$ (Appendix B). Additive inflation does not try to represent the

model error explicitly, but rather provides a lower bound for the forecast-error variances, thus preventing filter divergence, whereby the estimated forecast-error covariance $\mathbf{P}_e^{\mathrm{f}}$ becomes progressively smaller, leading the EnKF to give increasingly less weight to observations and allowing the forecast to drift further from reality. Moreover, the addition of random Gaussian perturbations can moderate the non-Gaussian higher moments that nonlinear error growth may have generated in the forecast step. Since the optimal EnKF solution assumes Gaussian distributions, this is expected to benefit the quality of the analysis es-

timate (Houtekamer and Zhang, 2016). However, adding random Gaussian noise may also mask useful covariance information pertaining to the model dynamics.

**Adaptive inflation**

Two popular flavours of adaptive multiplicative inflation are the Relaxation To Prior Perturbations (RTPP, Zhang et al. (2004)) and Relaxation To Prior Spread (RTPS, Whitaker and Hamill (2012)) methods. In the RTPP method, the analysis perturbations

$\mathbf{X}^{\mathrm{a}}$ are relaxed back to the forecast perturbations $\mathbf{X}^{\mathrm{f}}$ independently at each analysis point as follows:

$$\mathbf{X}^{\mathrm{a}} \leftarrow (1-\alpha_{\mathrm{RTPP}})\mathbf{X}^{\mathrm{a}} + \alpha_{\mathrm{RTPP}}\mathbf{X}^{\mathrm{f}}, \qquad (12)$$

where $\alpha_{\mathrm{RTPP}} \in [0,1]$ is a tuneable parameter. We demonstrate in appendix A that a 'no-perturbation' EnKF with $\alpha_{\mathrm{RTPP}} = 0.5$ is equivalent to the DEnKF algorithm, and we have chosen to implement the latter in this way. This equivalence is also exploited





in the operational MOGREPS-G (Met Office Global and Regional Ensemble Prediction System - Global) Ensemble of Data
Assimilations at the Met Office (Inverarity et al., 2022).

In our experiments, we also adopt the RTPS method – a purely multiplicative form of inflation – to inflate the spread of the
ensemble during each analysis step. In the RTPS method, the analysis ensemble spread $\sigma^{\mathrm{a}}$ is relaxed back to the forecast spread
$\sigma^{\mathrm{f}}$ by recomputing the analysis perturbation matrix $\mathbf{X}^{\mathrm{a}}$ as follows:

$$\mathbf{X}^{\mathrm{a}} \leftarrow \left(1 - \alpha_{\mathrm{RTPS}} + \alpha_{\mathrm{RTPS}} \frac{\sigma^{\mathrm{f}}}{\sigma^{\mathrm{a}}}\right) \mathbf{X}^{\mathrm{a}}. \tag{13}$$

where $\sigma$ is the spread at each gridpoint, measured as the square root of the diagonal entries of Eq. (7) applied to both the
forecast and analysis ensembles, and $\alpha_{\mathrm{RTPS}} \in [0,1]$ is a tuneable parameter.

### 3.3.3 Self-exclusion

Houtekamer and Mitchell (1998) identified the issue of inbreeding that occurs when ensemble perturbations from the ensemble
mean used to calculate $\mathbf{P}_{\mathrm{e}}^{\mathrm{f}}$ are themselves updated using this covariance matrix, leading to the spread being underestimated
(Roth et al., 2017). They adopted the approach of using two sub-ensembles and updating each sub-ensemble using forecast-
error covariances calculated from the other sub-ensemble and proposed the limit of calculating $\mathbf{P}_{\mathrm{e}}^{\mathrm{f}}$ using all members except the
one being updated. In our experiments, we have adopted this latter approach (called 'self-exclusion' by Bowler et al. (2017)).

## 4  Experiment setup and diagnostics

### 4.1  Initial conditions, boundary conditions and bottom topography

Motivated by the experiments with topography in Kent (2016) and Kent et al. (2017), non-rotating supercritical flow over
topography is considered for the experiments herein with non-dimensional parameter Fr = 1.1. The topography is defined as a
superposition of sinusoids in a sub-domain and zero elsewhere, i.e.

$$b(x) = \begin{cases} \sum_{i=1}^{3} b_i, & \text{for } x_p < x < x_p + 0.5; \\ 0, & \text{elsewhere;} \end{cases} \tag{14a}$$

$$\text{with } b_i = A_i \left(1 + \cos\left(2\pi(k_i(x - x_p) - 0.5)\right)\right), \tag{14b}$$

where $x_p = 0.1$, $k = \{2, 4, 6\}$, $A = \{0.1, 0.05, 0.1\}$. Given a non-zero initial velocity and periodic boundary conditions, this
collection of hills (see top panel in Fig. 2) generates varied and complex dynamics (including gravity-wave excitation) without
the need for external forcing or an imposed mean wind field. Periodic boundary conditions mean that waves that leave the
domain wrap around again, and so the flow remains energetic; this keeps the flow dynamically interesting without further
forcing. The basic (unperturbed) initial conditions used in these experiments are:

$h(x,0) + b(x) = 1; \quad hu(x,0) = 1; \quad hv(x,0) = 0; \quad hr(x,0) = 0;$  (15)



noting that since we consider non-rotating flow (effectively setting $\mathrm{Ro} = \infty$) over topography with transverse velocity $v$ initially zero, Eq. (4c) is removed from the integration.

For a given Froude number, potential characteristic scales of the dynamics can be analysed and, where possible, likened to high-resolution NWP. Consider a fixed length of domain $L_0 = 500$ km and velocity-scale $V_0 = 20$ ms$^{-1}$, implying a time-

scale $T_0 = 25000$ seconds ($\sim 6.94$ hours). Thus, one hour is equal to $0.144$ non-dimensional time units. A Froude number $\mathrm{Fr} = 1.1$ implies $gH_0 \sim 330$ m$^2$s$^{-2}$. We note that gravity $g$ in shallow water models can be reinterpreted as a reduced gravity $g' = g\Delta\rho/\rho$, where $\Delta\rho$ is the density difference between two fluid layers (see, e.g., Rogerson (1999)), and impose a height scale of $H_0 = 500$ m. This implies a reduced gravitational acceleration of $g' \sim 0.66$ ms$^{-2}$ and is justified as follows. Taking the average air density in the layers 0-500 m and 500-1500 m from the air density profile of the International Standard Atmo-

sphere[5], $\rho_{0-500\mathrm{m}} = 1.196$ kg m$^{-3}$ and $\rho_{500-1500\mathrm{m}} = 1.112$ kg m$^{-3}$, gives a reduced gravity of $g' \sim 0.69$ ms$^{-2}$. The threshold height $H_c$ mimics the Level of Free Convection (LFC, the height at which an air parcel lifted adiabatically, taking saturation into account, becomes warmer than its surroundings and starts to rise freely); we note here that this can be on the order of several hundred metres in convectively unstable conditions, so our choice of magnitude for $H_c$ and $H_r$ (Table 2) is reasonable. The initial ensemble should ideally be constructed to represent as fully as possible (given the finite ensemble size) the er-

ror statistics of the model state (Evensen, 2009). For convective-scale DA (and especially idealised experiments), spatially uncorrelated random Gaussian perturbations can be used to initialise the first cycle (e.g., Zhang et al. (2004)). A range of standard deviations $\boldsymbol{\sigma}^{\mathrm{ic}} = (\sigma_h^{\mathrm{ic}}, \sigma_{hu}^{\mathrm{ic}}, \sigma_{hr}^{\mathrm{ic}})$ used to sample zero-mean Gaussian initial condition perturbations have been trialed and, as noted by Houtekamer and Zhang (2016), the initial perturbations are forgotten promptly and negligible difference is noted between the trials after a few cycles. The perturbations used to generate the initial ensemble for all experiments are

$\boldsymbol{\sigma}^{\mathrm{ic}} = (0.1, 0.05, 0)$, i.e., for $j = 1, ..., N$:

$$h_j(x, 0) = h(x, 0) + \sigma_h^{\mathrm{ic}} \mathbf{z}_j, \quad \text{where } \mathbf{z}_j \sim \mathcal{N}(0, \mathbf{I}), \tag{16}$$

$\mathbf{I}$ is the 200-by-200 identity matrix, and $h(x, 0)$ is given in Eq. (15); initial ensembles for $hu$ and $hr$ are constructed analogously. Note that the rain variable is not perturbed ($\sigma_{hr}^{\mathrm{ic}} = 0$) since the rain field is initially zero everywhere and adding Gaussian noise is neither desired (producing unphysical negative rain) nor required (perturbations to the $h$ field lead to a random sample of

rain fields for $t > 0$).

Since the Kalman filter and its variants (and indeed most operational DA algorithms) are in essence Bayesian estimators that assume Gaussian statistics, the analysis update may produce a negative value for a state variable which should be strictly non-negative, e.g., rain rate or humidity. Numerical integration schemes that preserve non-negativity guaranteeing this is not the case in the forecast step, but spurious negative values may still result from the analysis step. For this idealised model, the

height and rain variables $h$ and $r$ should remain non-negative, with the numerics described in Kent et al. (2017) ensuring this in the forecast step. Negative $h$ is not only unphysical but also causes the subsequent integration to fail (by violating hyperbolicity of the model); negative $r$ poses no problems for the model integration but is clearly unphysical and impacts the other variables via the momentum coupling.

---

[5]See, for example: https://doi.org/10.1002/9781118568101.app2



The most straightforward solution is to enforce non-negativity simply by setting any spurious negative values to zero after the update. Whilst effectively ensuring the desired non-negative analysis states, this artificial modification is a crude approach that violates conservation of mass and may cause an 'initialisation shock' in the subsequent forecasts. More-sophisticated methods exist which incorporate constraints in the assimilation algorithm itself (e.g., Janjić et al. (2014)). However, these methods typically require a variational algorithm whose cost function includes terms to penalise unphysical negative solutions. We have instead adopted the non-variational Kalman gain formulation. In the idealised experiments presented here, any negative $h$ and $r$ values after assimilation are reset to a small value and zero, respectively, before the next forecast step.

## 4.2 Resolution, observing system and ensemble size

Current high-resolution NWP models are operating with a horizontal grid spacing on the order of one kilometre. For example, the Met Office's UKV model uses 1.5 km in its interior domain and the MOGREPS-UK (Met Office Global and Regional Ensemble Prediction System - United Kingdom) ensemble runs at 2.2 km (Tang et al., 2013; Hagelin et al., 2017); the Deutscher Wetterdienst's ICON-D2 model has a 2.2 km horizontal grid spacing[6]. Running models at these resolutions means that some aspects of convection are resolved explicitly (albeit imperfectly) and this yields precipitation fields that look more realistic (Lean et al., 2008). With this in mind, a forecast grid spacing of 2.5 km is imposed for the idealised model. Thus, given a length $L_0 = 500$ km of the domain, the computational grid for the forecast model has $n_x = 200$ grid points and the total number of degrees of freedom is $n = 600$ (again noting that $hv$ is removed from the integration since the flow is non-rotating). Observations are assimilated hourly, comparable with the cycling frequency of operational convective-scale systems (e.g., Milan et al. (2020)), over a 48 hour period. All variables are observed directly (making the observation operator $\mathbf{H}$ linear) with specified random error standard deviations $\sigma = (\sigma_h, \sigma_u, \sigma_r)$, used to generate synthetic observations, and spatial density of an observation every 25 grid points ($\sim 63$ km) for $h$ and 20 grid points ($\sim 50$ km) for $u$ and $r$. This means that $p = 28$ observations are assimilated at each analysis step. To ensure that our idealised system exhibits rank–deficiency akin to operational systems, i.e., $N < p < n$, we employ an ensemble with $N = 18$ members. This relatively low number of observations is further motivation for our choice of the DEnKF over the stochastic 'perturbed-observation' EnKF. The DEnKF avoids introducing further sampling error in representing the observation-error covariance matrix $\mathbf{R}$ that would otherwise result from perturbing the small number of observations, which can in turn make it more likely that the state vector has negative $r$ or from the fluid depth being on the wrong side of a convection or precipitation threshold.

Similarly to the approach of the previous section, we have applied an additional quality control step of resetting any unphysical negative $h$ and $r$ synthetic observations to 0.001 and 0, respectively. Observation quality control is a crucial part of operational NWP; in an idealised setting, it can also help to reduce forecast failures and enables a larger range of the parameter space to be explored.

---

[6]See the webpage at https://www.dwd.de/EN/ourservices/nwp_forecast_data/nwp_forecast_data.html





## 4.3 Diagnostics for assessing performance and relevance

### 4.3.1 Ensemble error and spread

The performance of ensemble methods is typically measured by their accuracy, quantified by the RMSE of the ensemble mean and ensemble spread (e.g., Whitaker and Loughe (1998)). A useful summary measure for our dimensionless system is provided by the RMSE of the ensemble mean, defined as:

$$\text{RMSE} = \sqrt{\frac{1}{n}\sum_{k=1}^{n}(\overline{x}_k - x_k^{\text{nat}})^2}, \quad \text{with} \quad \overline{x}_k = \frac{1}{N}\sum_{j=1}^{N} x_{j,k},\tag{17}$$

where $x_k^{\text{nat}}$ and $x_{j,k}$ are the $k^{\text{th}}$ components of the nature run vector $\mathbf{x}^{\text{nat}}$ and ensemble member $\mathbf{x}_j$, respectively. A natural measure of the typical spread (or dispersion) of the ensemble is provided by the root mean value of the trace Tr of $\mathbf{P}_e^{\text{f}}$, given by the sum of its diagonal entries:

$$\text{SPR} = \sqrt{\frac{1}{n}\sum_{k=1}^{n}\frac{1}{N-1}\sum_{j=1}^{N}(x_{j,k} - \overline{x}_k)^2} \equiv \sqrt{\frac{1}{n}\text{Tr}(\mathbf{P}_\text{e}^\text{f})}.\tag{18}$$

To ensure that each variable is of the same order of magnitude, and since $r \sim \mathcal{O}(0.01)$ but $h$ and $u \sim \mathcal{O}(1)$, we will multiply $r$
by 100 before computing these two statistics.

Both RMSE and SPR are valid at the same time as the state vector. As diagnostics, we monitor the both SPR and RMSE individually, and also the ratio SPR/RMSE. An ideal ensemble's spread is expected to match the RMSE of its mean at the same lead time in order to represent the full uncertainty in the forecast (Stephenson and Doblas-Reyes, 2000). However, Leutbecher and Palmer (2008) showed that, for a finite ensemble of $N$ members, the ensemble spread converges to the RMSE if
a correction factor $\frac{N+1}{N-1}$ is applied[7]. Although a ratio of one is considered to be optimal, in practice this ratio fluctuates in time and space, as well as between variables, and so we accept the ratio within a certain tolerance (e.g., SPR/RMSE = $1 \pm 0.2$). This tolerance also takes into account the bias correction described above (Leutbecher and Palmer, 2008). These measures provide a simple but important objective diagnostic on the reliability of the generated ensemble in the EnKF (cf. Piccolo et al. (2019)).

### 4.3.2 Continuous Ranked Probability Score

A well-configured ensemble is crucial to good DA performance as it is used to estimate flow–dependent forecast–error covariances. Spread and RMSE are a good first check for an adequately performing ensemble, but do not capture the entire permissible range of outcomes, i.e., the full distribution provided by the ensemble. The CRPS verifies the reliability of an ensemble for scalar quantities, and is a popular verification tool for probabilistic forecasts. Reliability represents the degree
to which forecast probabilities agree with outcome frequencies. A key feature of the CRPS is that it generalises the mean

---

[7]Leutbecher and Palmer (2008) define spread by dividing squared differences by N rather than N-1. Adopting the latter definition changes their correction factor to (N+1)/N.





absolute error to which it reduces for a deterministic forecast. It is a negatively oriented scoring rule that assigns a numerical score (lower scores indicating higher skill with zero being perfect) to probabilistic forecasts. The CRPS is computed for each component of the forecast/analysis ensemble $\mathbf{x}_j^{\mathrm{f/a}}$ at every assimilation time, using the formulation of Hersbach (2000).

### 4.3.3 Observational Influence Diagnostic

The global average observation influence diagnostic, related to the Degrees of Freedom for Signal measure, is defined as:

$$\mathrm{OID} = \frac{\mathrm{Tr}(\mathbf{S})}{p} \tag{19}$$

where $\mathbf{S} = \partial(\mathbf{H}\mathbf{x}^{\mathrm{a}})/\partial\mathbf{y} = \mathbf{H}\mathbf{K}$ is the analysis sensitivity matrix (Cardinali et al. (2004); see appendix D for a derivation generalised for an ensemble system with self-exclusion). It provides a norm for quantifying the overall influence of observations on the analysis estimate expressed in observation space and is normalised by the total number of observations $p$ to express the

result as a fraction, with 1 representing an analysis totally determined by the observations and 0 an analysis that is equal to the prior forecast. Thus, increasing the number of observations alone will not necessarily increase their overall influence. Cardinali et al. (2004) have applied this diagnostic to the ECMWF global NWP model and found an observation influence $\mathrm{OID} = 0.18$. This means that observations adjust the prior forecast estimate closer to reality, rather than replace it completely; observations alone are too few and incomplete (compared to the size of the system) to provide a comprehensive picture of the state. However,

it should be noted that the prior forecast estimate contains observational information from previous analysis cycles. Brousseau et al. (2014) used the same diagnostic to quantify the observational impact for the AROME (Applications de la Recherche l'Opérationnel à Méso-Echelle) convective-scale forecast-assimilation system and quote values between $0.25 - 0.4$ for two-meter temperature and between $0.6 - 0.7$ for two-meter relative humidity and 10m wind. It is expected that the observational influence for convective-scale NWP is higher than for global NWP, but that the analysis estimate is still dominated by the prior

forecast.

Therefore, we consider our idealised system relevant for convective-scale NWP if it has an influence $20\% \lessapprox \mathrm{OID} \lessapprox 40\%$. The OID is computed at each assimilation time and is also expected to vary temporally; it can also be readily partitioned to assess observations of each variable. Throughout this work, we have calculated $\mathrm{OID}_j$ for each ensemble member $j$, taking account of self-exclusion, and averaged these $N$ values to provide a single measure of the observation influence on the ensemble

as a whole (Appendix D).

### 4.3.4 Error growth rates

The error–doubling time $T_{\mathrm{d}}$ of a forecast-assimilation system is the time taken for the error $E$ of a finite perturbation at time $T_0$ to double:

$$\frac{E(T_{\mathrm{d}})}{E(T_0)} = 2, \tag{20}$$

where $E$ is some error norm (taken here to be the RMSE of the differences between each member and the nature run, rather than just using the ensemble mean). The error–doubling time is expected to fluctuate somewhat between variables (since





certain variables behave more nonlinearly than others) and is controlled by the 'dynamics of the day', i.e., it is temporally and spatially dependent. As such, it is not a fixed number, but by averaging over a number of staggered forecasts covering a range of dynamics and initial perturbations, the mean error–doubling time of the system can be estimated. For global NWP, Buizza

(2010) found a doubling time of 1.28 days for the Northern Hemisphere forecast error. Errors in high-resolution NWP grow faster at the smallest scales than in the global case owing to the strong nonlinearities at convective scales. Thus, in order to be relevant for convective-scale NWP, a well-tuned idealised forecast–assimilation system should exhibit a mean error–doubling time on the order of hours rather than a day. Moist convection severely limits mesoscale predictability (Zhang et al., 2003), and for limited–area cloud–resolving models, the mean error–doubling time has been found to be around 4 hours (Hohenegger and

Schär, 2007). Thus, ensemble forecasts initialised with the analysis perturbations from a well-tuned experiment should exhibit characteristic error growth rates on this timescale. This diagnostic is not used in the tuning process but is used to ascertain the relevance of an experiment considered well-tuned with respect to the three criteria of Table 1. We compute $T_d$ for an idealised ensemble prediction system in section 5.2 by running $450$ $24$ hr forecasts initialised with the analysis ensembles from $25$ assimilation cycles (noting that 18 members times 25 cycles equals 450 initial conditions for the forecasts).

### 445   4.3.5   Subjective verification

The objective measures detailed above are complemented by 'subjective verification' at a later stage of assessment, in order to give a comprehensive understanding of system performance and relevance. Subjective visual evaluation of individual experiments is recognised as playing an important role in the verification process of model and assimilation performance (e.g., Piccolo et al. (2019)), particularly for convection-permitting models and precipitation forecasts. Operational centres use

subjective verification, in tandem with more-traditional objective statistical measures, to comprehensively assess changes to operational forecast-assimilation systems (e.g., Rawlins et al. (2007); Lean et al. (2008)). Unlike objective measures, subjective verification techniques require both knowledge of meteorology and modelling, as well as experience accrued in assessing previous outputs (Mittermaier et al., 2016). Careful interpretation at this subjective level may provide crucial insight that can not be achieved via objective verification alone. However, subjective verification is time-consuming and not systematic. We

therefore examine model outputs after the objective measures have identified credible experiments, in order to substantiate and justify the claim that an experiment is both well-tuned and relevant.

## 5   Experiment results

### 5.1   Tuning and assessment of filter performance

In this section, we outline the tuning process and give a first assessment of filter performance for a range of experiments.

Despite the use of a low-dimensional, inexpensive model, the tuning of this idealised forecast-assimilation system requires many adjustments to most of the parameters of the observing system and filter configuration (see right column of Table 2). The observing system parameters are the observing frequency, total number of observations, their spatial density, and error;

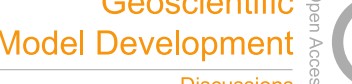

**Table 2.** An overview of the model and assimilation parameters used in the idealised experiments. Units are dimensionless unless stated otherwise. The $L_{\mathrm{loc}}$ values correspond to cut-off distances $(2c)$ of 1000 km, 500 km, $333\frac{1}{3}$ km, and 250 km, respectively.

| Model parameters | | Fixed parameters | |
|---|---|---|---|
| Rossby number, Ro | $\infty$ | Update frequency [min] | 60 |
| Froude number, Fr | 1.1 | Forecast $n_x$ | 200 |
| Convection threshold, $H_c$ | 1.02 | Nature $n_x^{\mathrm{nat}}$ | 400 |
| Rain threshold, $H_r$ | 1.05 | Ensemble size $N$ | 18 |
| $\alpha$ | 10 | Number of observations $p$ | 28 |
| $\beta$ | 0.2 | Obs. density $(d_h, d_u, d_r)$ [km] | (63, 50, 50) |
| $c_0^2$ | 0.085 | Obs. error $(\sigma_h, \sigma_u, \sigma_r)$ | (0.05, 0.02, 0.003) |
| Initial conditions | Eq. (15) | Adaptive inflation, $\alpha_{\mathrm{RTPP}}$ | 0.5 |
| Boundary conditions | Periodic | Tuning parameters | |
| Domain length $L_0$ [km] | 500 | Localisation scale, $L_{\mathrm{loc}}$ | {0.5, 1, 1.5, 2} |
| Velocity scale $V_0$ [ms$^{-1}$] | 20 | Adaptive inflation: $\alpha_{\mathrm{RTPS}}$ | {0.1, 0.3, 0.5, 0.7, 0.9} |
| Height scale $H_0$ [m] | 500 | Additive inflation $\gamma_{\mathrm{a}}$ | {0.05, 0.08, 0.1, 0.12, 0.15, 0.2, 0.3, 0.4, 0.5} |

filter configuration parameters are ensemble size $N$, localisation length-scale $L_{\mathrm{loc}}$ and the inflation factors, $\alpha_{\mathrm{RTPS}}$ and $\gamma_{\mathrm{a}}$. In particular, the opportunity (and indeed need) to vary the observing system greatly increases the dimensionality of the overall

tuning process. For the sake of simplicity, though, we restrict our considerations here to the tuning of the filter configuration given a particular observing system and ensemble size. Therefore, given the model configuration described in section 3 (and summarised in the left column of Table 2) and a prescribed observing system, each experiment is defined by a combination of the tuning parameters; these experiments are then systematically compared with each other and assessed via the diagnostics presented in section 4.3 in pursuit of a well-tuned experiment.

Before asking how well a filter performs (and before considering the effect of both localisation and inflation), we need to choose the timescale for which the system will be optimised. In particular, since our focus is convective-scale NWP nowcasting, we optimise the system for forecasts with a three-hour lead time. It is important to note that there is not a single most 'well-tuned' configuration but rather a subset of the parameter space that yields a selection of well-tuned experiments. Our tuning process selects those experiments that satisfy the tuning criteria of Table 1; once this has been achieved, candidate experiments

are validated further for relevance (section 5.2). Thus, we focus now on the filter diagnostics with the following targets:

(i)   SPR/RMSE $\sim 1 \pm 0.2$,

(ii)   minimum value of RMSE,

(iii)   minimum value of CRPS.







**Figure 5.** Summarising the tuning process: SPR/RMSE (top-left), OID (top-right), RMSE of the ensemble mean (bottom-left), and CRPS (bottom-right) as a function of the tuning parameters $L_{\mathrm{loc}}$, $\gamma_{\mathrm{a}}$, and $\alpha_{\mathrm{RTPS}}$. Spread, error and CRPS values are computed on forecasts with a 3 hr lead time and averaged over space, time, and model variables; the OID is computed at each assimilation time and then averaged over time and model variables. For all panels, light/white shading indicates desired values. Top-left: squares outlined in red identify experiments with SPR/RMSE close to 1. Top-right: the OID (expressed as a percentage). Bottom-left and -right: for those experiments already highlighted in the top-left figure, the squares outlined in black denote experiments with minimal RMSE and CRPS, respectively.

For each of the above targets, space- and time-averaged statistics are computed for each experiment (after omitting the first 12 hours in which the experiments are spinning up), and these values are then averaged over the three model variables $h$, $u$, and $100r$ (section 4.3.1). The benefit of doing so means that each experiment is assigned a single number per diagnostic; this enables a straightforward and objective comparison between experiments, as summarised in Fig. 5. Each sub-panel comprises 45 cells, each giving the value of the diagnostic as a function of the inflation factors ($\alpha_{\mathrm{RTPS}}$ and $\gamma_{\mathrm{a}}$) and the localisation length-scale $L_{\mathrm{loc}}$. The target value for a well-tuned experiment is indicated by the white/light cells. By comparing the position of the





best experiments under each criterion, we seek a region of the parameter space that satisfies all the conditions above. However, as is often the case in tuning a complex system, detecting such a subset of experiments is non-trivial and consensus is hard to achieve. As a compromise, we set a tolerance of $\pm 0.2$ on the condition SPR/RMSE $\sim 1$ and demand that the experiments must first satisfy this target in order to be evaluated for minimal RMSE and CRPS: these experiments are outlined in red in the top-left panel of Fig. 5. Among them, (some of) those showing low values of RMSE and CRPS are outlined in black in the bottom-left and bottom-right panels, respectively.

The SPR/RMSE measure and CRPS assess the filter configuration of the forecast-assimilation system, in particular the role of the ensemble, but they do not entirely indicate its relevance to the NWP problem (in particular the observing system). To this end, we also show the same graphic for the OID (top-right), which has a target value $20\% \lessgtr \text{OID} \lessgtr 40\%$ indicated by and centred on white/light shading. Given that the imposed observation error is fixed for these experiments, as well as the update frequency and observation density, the subsequent influence of the observations is controlled by the changing impact of the forecast (due to inflation and localisation). The average influence of the observations increases with increasing inflation, as should be expected: higher inflation factors increase the ensemble spread, and consequently lead to larger variances in the forecast-error covariance matrix $\mathbf{P}_e^{\text{f}}$, representing decreased confidence in the accuracy of the prior forecast. It follows that more weight is given to the observations in the filter and increases their influence on the analysis estimate. Crucially, there is good agreement between the desired SPR/RMSE and OID, indicating at this stage of the assessment that a good filter configuration and meaningful observing system can be achieved within this experimental set-up; this is examined further in section 5.2.

As a consequence of the application of the three tuning criteria above, we focus now on 12 candidate well-tuned experiments, indicated by the squares outlined in black in Fig. 5, and examine the role of localisation. Figure 6 shows the forecast-error correlation matrices (whose entry in the $i^{\text{th}}$ row and $j^{\text{th}}$ column is $\left(\mathbf{P}_e^{\text{f}}\right)_{i,j} / \sqrt{\left(\mathbf{P}_e^{\text{f}}\right)_{i,i} \left(\mathbf{P}_e^{\text{f}}\right)_{j,j}}$) of four different experiments, all drawn from the 12 selected above, one for each $L_{\text{loc}}$ value, before and after the application of localisation. Although the impact on the experiments of different localisation scales is not entirely evident in Figure 5, here the direct effect on the forecast-error correlation matrix is much clearer. In particular it is clear that, on the one hand, for values of $L_{\text{loc}}$ bigger than 1 (bottom panels), much of the signal is suppressed away from the diagonal bands, leaving in place only the narrow diagonal band-like structures present in the matrices. On the other hand, a value of $L_{\text{loc}} = 0.5$ (top-left panel) is too broad to remove spurious long-distance correlations in $\mathbf{P}_e^{\text{f}}$. Since the aim of localisation is to remove the spurious long-distance correlations while retaining valid correlations, and since the typical horizontal length-scale in operational convective-scale NWP (see Table 2) is on the order of 100 km, we focus now on the three experiments with $L_{\text{loc}} = 1.0$, recalling that this defines a dimensionless length-scale of $2c = 1.0$ (i.e., correlations are suppressed completely beyond 500 km).

Another key aspect of verification that has not been properly scrutinised yet is to assess whether the data assimilation is actually improving forecast accuracy (and by how much). This is ascertained by comparing forecasts with different lead times, but valid at the same time, in an attempt to highlight the benefit that assimilating new observations had on the most recently initialised forecast. Figure 7 shows, for the three highlighted experiments with $L_{\text{loc}} = 1.0$ in Fig. 5, the domain-averaged time series of ensemble SPR and RMSE of the ensemble mean for each field $h$, $u$ and $r$. These values are computed from 3 hr





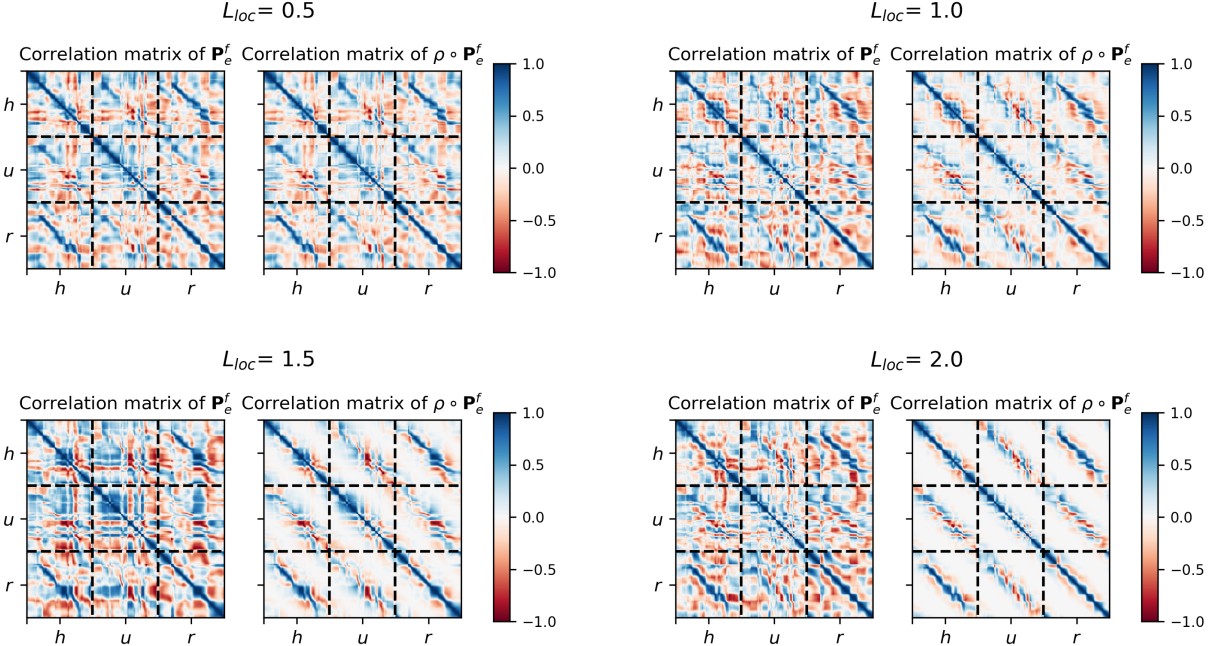

**Figure 6.** Examining the impact of localisation lengthscales: raw and filtered forecast-error correlation matrices (i.e., normalised $\mathbf{P}_e^f$) for $L_{\mathrm{loc}} = 0.5, 1.0, 1.5, 2.0$ (clockwise from top-left), with the $L_{\mathrm{loc}}$ values corresponding to cut-off distances ($2c$) of 1000 km, 500 km, $333\frac{1}{3}$ km, and 250 km, respectively. The matrices are constructed from forecast perturbations valid at $T = 36$ hrs; note that each matrix is the average over the $N$ different correlation matrices that arise due to self-exclusion (see step 2iii in appendix B). The following inflation factors are used in each experiment: $\gamma_a = 0.08$, $\alpha_{\mathrm{RTPS}} = 0.7$ (top left), $\gamma_a = 0.15$, $\alpha_{\mathrm{RTPS}} = 0.7$ (top right), $\gamma_a = 0.12$, $\alpha_{\mathrm{RTPS}} = 0.7$ (bottom left), $\gamma_a = 0.12$, $\alpha_{\mathrm{RTPS}} = 0.7$ (bottom right).

(blue lines) and 4 hr (red lines) forecasts valid at the same time; the average over these times (excluding the first 12 hours for spin-up) for each variable is shown in the top-left corner of each panel, and detailed further in Table 3. Examining forecast errors exemplifies the impact of data assimilation: we expect the 3 hr forecast to be more accurate (i.e., lower RMSE values) than the 4 hr forecast, since the former has been updated with more-recent observational information than the latter. Indeed, this is evident in all three experiments, with the 3 hr forecast error (blue dashed lines) generally lower than the 4 hr forecast error

(red dashed lines) throughout the 48 hours. This is confirmed quantitatively in time-averaged values for all variables (Table 3). The 3 hr forecasts are consistently more accurate than the 4 hr forecasts, with the percentage improvement ranging between 5–12% between variables, and close to 10% overall. Interestingly, the precipitation field exhibits the largest improvement out of the model variables in all three experiments.

Furthermore, fluctuations in these time series highlight the time-dependence of the ensemble SPR and consequently the

flow-dependence of forecast error. We note their quasi-oscillatory nature, which is attributed to the periodic boundary conditions. In general, SPR (solid lines) is comparable with the RMSE of the ensemble mean (dashed lines), indicating that the



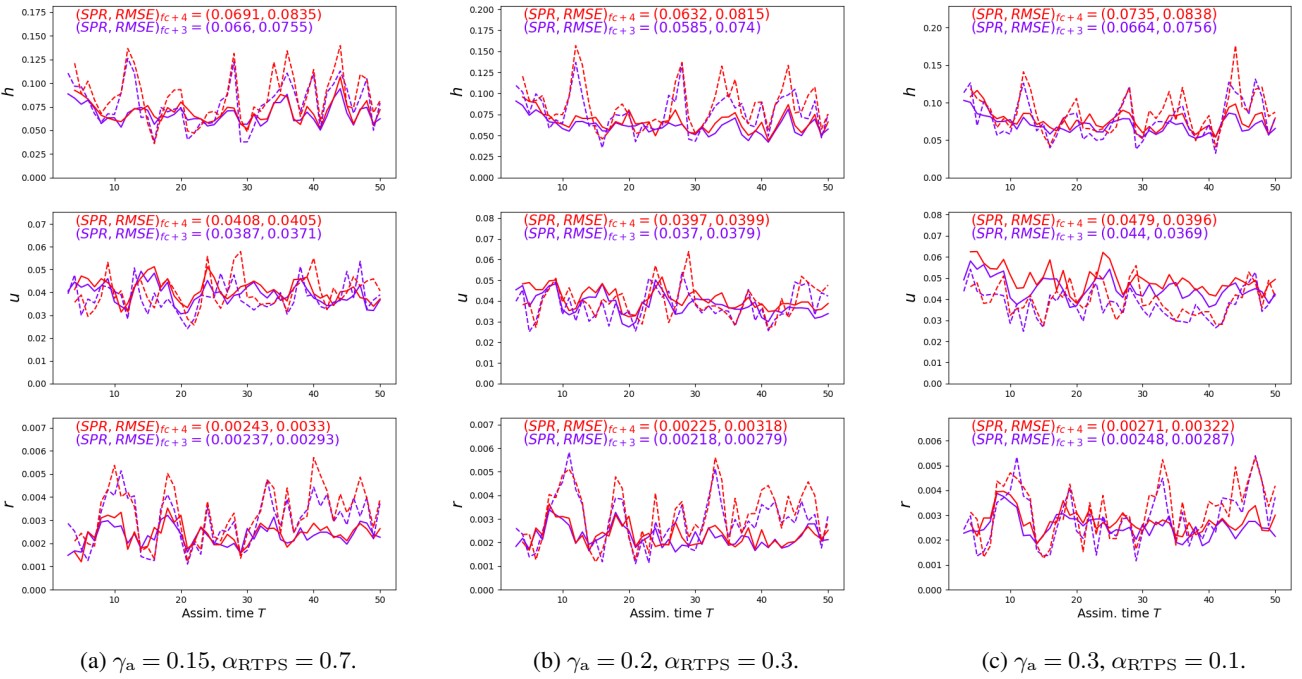

**Figure 7.** Domain-averaged RMSE of the ensemble mean (dashed lines) and SPR (solid lines) for the 4 hr (red) and the 3 hr (blue) forecast ensembles: $h$ (top), $u$ (middle), and $r$ (bottom), with time-averaged values (excluding the first 12 hrs) given in the top-left corner of each panel and analysed further in Table 3. The parameters that define each experiment are given in the subcaptions. All experiments have $L_{\mathrm{loc}} = 1.0$.

| Variable | Experiment (a): $\gamma_{\mathrm{a}} = 0.15$, $\alpha_{\mathrm{RTPS}} = 0.7$ | | | Experiment (b): $\gamma_{\mathrm{a}} = 0.2$, $\alpha_{\mathrm{RTPS}} = 0.3$ | | | Experiment (c): $\gamma_{\mathrm{a}} = 0.3$, $\alpha_{\mathrm{RTPS}} = 0.1$ | | |
|---|---|---|---|---|---|---|---|---|---|
| | 3 hr | 4 hr | % diff | 3 hr | 4 hr | % diff | 3 hr | 4 hr | % diff |
| $h$ | 0.0755 | 0.0835 | 9.6% | 0.0740 | 0.0815 | 9.2% | 0.0756 | 0.0838 | 9.8% |
| $u$ | 0.0371 | 0.0405 | 8.4% | 0.0379 | 0.0399 | 5.0% | 0.0369 | 0.0396 | 6.8% |
| $r$ | 0.00293 | 0.00330 | 11.2% | 0.00279 | 0.00318 | 12.2% | 0.00287 | 0.00322 | 10.9% |
| Average | | | 9.7% | | | 8.8% | | | 9.2% |

**Table 3.** RMSE values for the three experiments in Figure 7 (all with $L_{\mathrm{loc}} = 1.0$): comparison of 3 hr and 4 hr forecast errors, and the percentage differences. The % difference is computed with respect to the 4 hr forecast: $(\mathrm{RMSE}_{T+4} - \mathrm{RMSE}_{T+3})/\mathrm{RMSE}_{T+4}$.

ensemble is providing an adequate estimation of the forecast-error covariance matrix $\mathbf{P}_e^{\mathrm{f}}$. There is naturally some variability between variables, and we note that $h$ and $r$ tend to be more underspread than $u$ (i.e., $\mathrm{SPR}/\mathrm{RMSE} \lesssim 1$ for $h$ and $r$, whereas $\mathrm{SPR}/\mathrm{RMSE} \sim 1$ for $u$). Dynamically this makes sense: the convective behaviour in the flow is driven by $h$ and manifested

in $r$. These variables exhibit highly nonlinear behaviour in regions of convection that the ensemble and filter (which assumes Gaussian statistics) struggle to capture in terms of SPR. We discuss this further in section 5.3. Also noteworthy is that these series are stationary in the sense that the spread and error do not drift in time. A poorly configured filter may diverge from 'real-





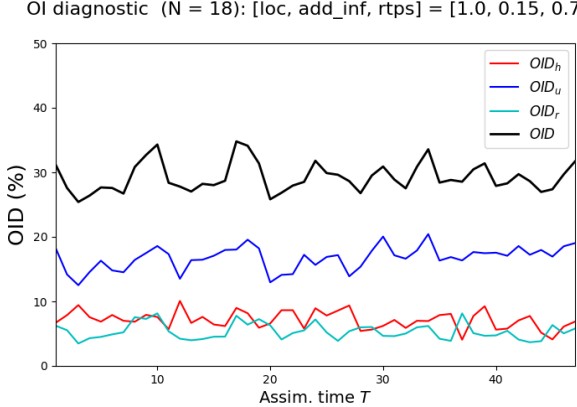

**Figure 8.** Time series of the observational influence diagnostic (%): the overall influence (thick black line) fluctuates around 30%. Coloured lines indicate the influence of the individual observation types that sum to the overall influence.

ity' (despite the frequent update from observations), but we see here that these set-ups do not experience this filter divergence. Of the three experiments examined in Fig. 7, we focus next on experiment (a), which has the largest overall improvement in
forecast accuracy (i.e., largest reduction of RMSE: see Table 3, left column). It also has the lowest additive inflation factor, which is desirable from a model-error perspective, and an RTPS value ($\alpha_{\mathrm{RTPS}} = 0.7$) comparable with operational systems (cf. Inverarity et al. (2022), who use a value of 0.8).

## 5.2 Validation and relevance for convective-scale NWP

The CRPS and SPR/RMSE measures ascertain the general performance of the forecast-assimilation system itself, in particular
the role of the ensemble, but they are not sufficient to indicate its relevance to the NWP problem. To this end, two additional validation diagnostics are employed:

(i) the observational influence diagnostic (OID);

(ii) error-doubling time statistics.

The OID is calculated at each cycle and is expected to vary for the given dynamical situation – the 'weather-of-the-hour'. If
at a given time there is a lot of uncertainty in the forecasts, e.g., due to a lot of convective behaviour and associated nonlinearity, then it is to be expected that the observations have a greater influence at this time. On the other hand, a situation without much convection is relatively predictable, suggesting more certainty in the forecasts and less impact from the observations. The OID of $h$, $u$, and $r$ is plotted in Fig. 8 as a function of time. The overall influence (thick black line) is typically $25-35\%$, comparable to operational convective-scale forecast–assimilation systems. The influence of $h$, $u$ and $r$ observations is also shown: this too
fluctuates depending on the 'hourly weather', and we note that the average influence of observing $u$ over 48 hours is higher than for $h$ and $r$ (which are themselves comparable). We do not claim that this is an intrinsic property of our model and recognise



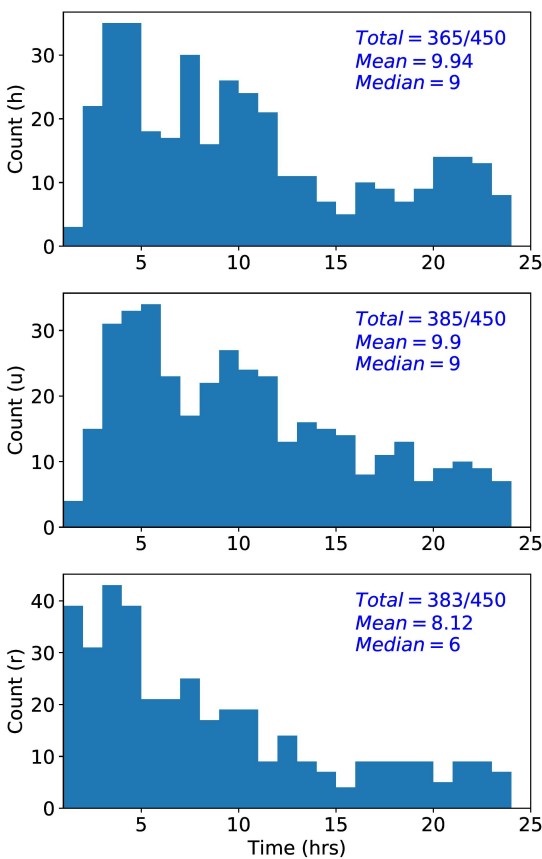

**Figure 9.** Histograms of the error-doubling times for 450 24-hour forecasts initialised using the analysis ensembles from the idealised forecast-assimilation system. From top to bottom: $h$, $u$, $r$. For each variable, the number of forecasts in which the initial error doubled within 24 hrs, as well as the mean and median of the time taken (in hours), are given in the top-right of each panel.

that choosing a different observing system will lead to a different balance between the various observational components. Monitoring the observational influence in this way facilitates further investigations of the observing system; this is however not the purpose of this study, but we conclude that this well-tuned experiment has an observing system that is relevant for operational convective-scale NWP (for which $20\% \lessapprox \text{OID} \lessapprox 40\%$).

The final objective verification measure of our protocol is the error-doubling time, described in section 4.3. The goal of data assimilation is to provide the best estimate of the state of the atmosphere by merging forecast and observational information. Typically, this best estimate is used to initialise forecasts that run longer than the length of the assimilation window. To provide further validation of the relevance for convective-scale NWP, the error–doubling time statistics are considered by running numerous forecasts initialised at different times with the analysis ensembles produced in this experiment. Each cycle provides an ensemble of $N = 18$ analyses and, by taking a range of initial conditions from successive cycles, the resulting forecasts



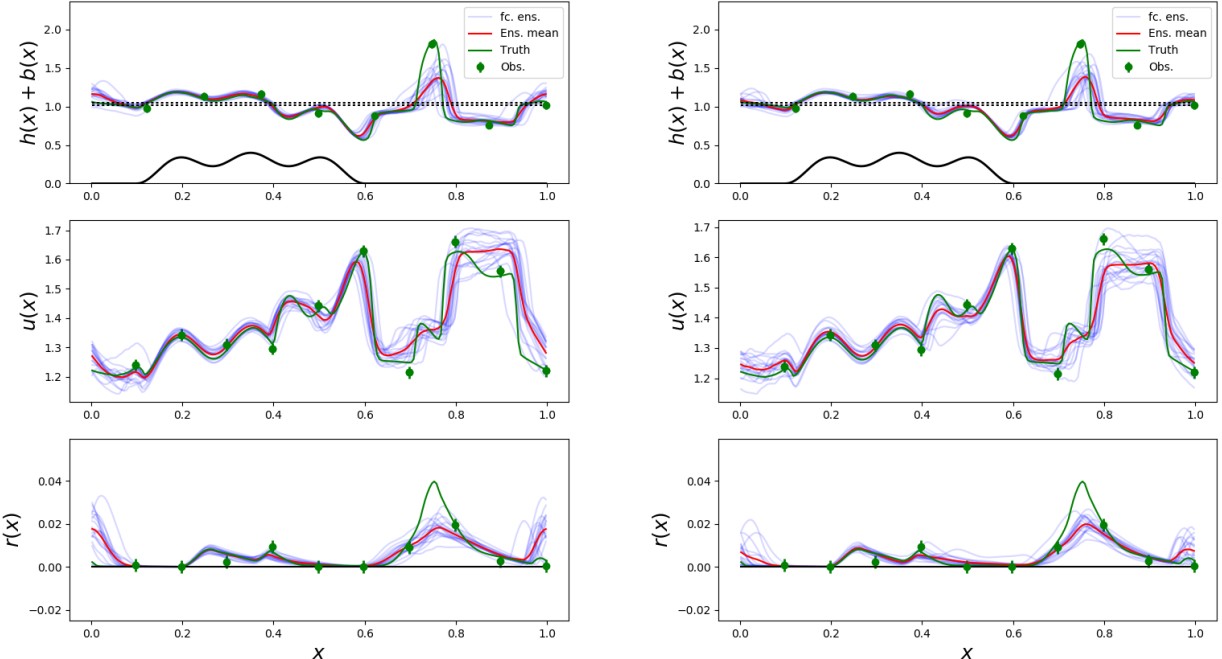

**Figure 10.** Ensemble trajectories (blue) and mean (red), location of synthetic observations (green circles with error bars, noting that different observation values are used at the initialisation times for each forecast), and nature run (green solid line, labelled as 'truth' in the legend) valid at $T = 40$ hrs. Left: 4 hr forecast (initialised at $T = 36$ hrs). Right: 3 hr forecast (initialised at $T = 37$ hrs). Unphysical negative h and r synthetic observations are reset to zero.

cover a wide range of dynamics. In total, 450 24-hour forecasts are run and the time $T_\mathrm{d}$ taken for the initial RMSE to double (Eq. (20)) is recorded: histograms of the error-doubling times for each variable are shown in Fig. 9. We note that $h$ and $u$ have similar doubling times (about 9 hrs), while errors in the precipitation field $r$ grow at a faster rate (doubling in about 6 hrs).

Dynamically, this is to be expected due to the highly nonlinear behaviour of $r$. As noted in section 4.3.4, the average doubling time in convection-permitting NWP models is around 4 hours (Hohenegger and Schär, 2007). It should also be noted that the resolution mismatch between forecast and nature runs strongly determines error-growth rates: experiments whose nature run resolution has been doubled ($n_x^\mathrm{nat} = 800$; cf. Fig. 2) have error-doubling times of around 4 hours (not shown), approximately half the values of this experiment.

**5.3 Subjective verification**

We conclude the assessment by presenting full–domain snapshots of the dynamics, drawn from the well-tuned experiment examined in the previous section (experiment (a) in Table 3) in Figs. 10 and 11. Individual ensemble members (blue lines), the ensemble mean (red lines for the forecasts and cyan lines for the analyses), synthetic observations (green dots), and the





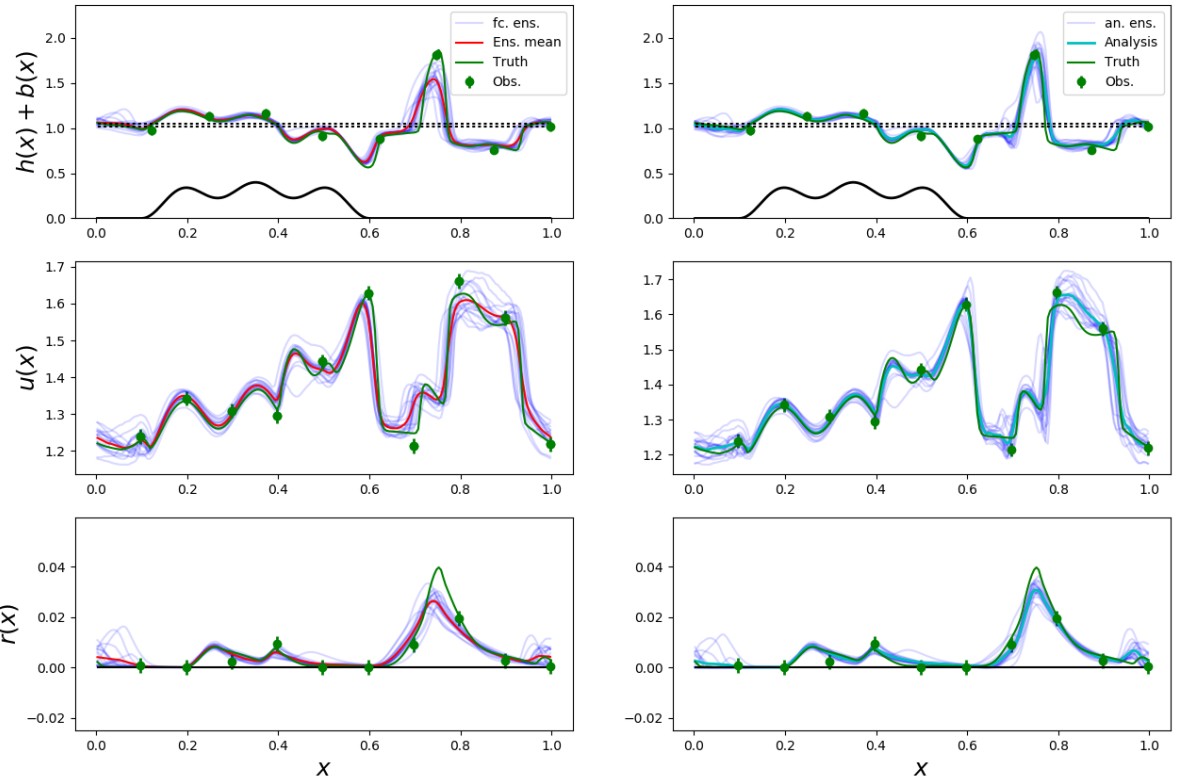

**Figure 11.** Ensemble trajectories (blue) and mean (red forecast; cyan analysis), synthetic observations (green circles with error bars, again assimilating different observation values in these forecasts), and nature run (green solid line) valid at $T = 40$ hrs. Left: (1 hr) forecast. Right: analysis.

verifying nature run (green lines) of each variable are displayed at a single illustrative time. Since the system's performance

has already been evaluated objectively, the purpose of this section is to demonstrate from a qualitative point of view the impact of data assimilation on the system, and to highlight both its successes and struggles in coping with the nonlinear behaviour of the model.

Figure 10 compares the 4 hr forecast (initialised at $T = 36$ hrs; left panel) with the 3 hr forecast (initialised at $T = 37$ hrs; right panel) valid at $T = 40$ hrs. Clearly, we expect the latter to be more accurate than the former (i.e., closer to the nature run

trajectory), since it benefits from a more-recent initialisation and therefore has had less time to diverge.

The dynamical situation (from the nature run; green lines) at $T = 40$ hrs is as follows. Since $u > 0$, the flow is moving from west (left) to east (right) with convection downstream of the topography. There is one active region of convection observable in $h$ around $x = 0.75$ with associated precipitation $r$ and also some precipitation over the topography $0.2 < x < 0.5$. Sharp gradients in wind $u$ are present where there are sharp gradients in fluid depth $h$; recall that precipitation forms when both

$h + b > H_r$ and $\partial_x u < 0$, and the rate of formation is proportional to $-\beta \partial_x u$. The fluid marginally exceeds the threshold





heights around $x = 1$ (or 0 recalling periodic boundaries). Note that the green circles denote the synthetic observations valid at $T = 40$ hrs; the synthetic observations assimilated at $T = 36$ hrs and $T = 37$ hrs to produce the forecast trajectories shown in Figure 10 have the same spatial locations but different values.

Generally, the forecasts are accurate (close to the nature run with small spread) for all three fields in non-convecting regions.
The two regions of less accuracy (and hence higher uncertainty exemplified by the spread) are the active convection around $x = 0.75$ and the region close to the thresholds around $x = 1$. Both the 3 hr and 4 hr forecasts capture the location of convection but struggle with its intensity, which we have already seen in Figure 2 is a consequence of using a lower-resolution forecast model. Somewhat unsatisfactorily, there is little noticeable difference between the two forecasts in $h$ and $r$, suggesting that the more-recently initialised forecast offers only marginal improvement in this case. Perhaps this is not unexpected, given the
rapid ($< 3$ hrs) development of convection here. Furthermore, given the limitations of the forecast model and the time since initialisation, it is not able to capture the intensity. In contrast, visual examination of the forecasts around $x = 0/1$ shows greater improvement. Both forecasts predict spurious precipitation, which is not present in the nature run, but the amount is greatly reduced in the 3 hr forecast. This is a particularly interesting situation offered by the model: some ensemble members exceed the thresholds while others do not. The model thus exhibits highly nonlinear behaviour that is an inherent characteristic of
the onset of convection and precipitation, reflected in the non-Gaussian distribution and large spread of the forecast ensemble. Despite the challenges posed by this behaviour and the limitations of the EnKF, the impact of the data assimilation is positive and leads to a more-accurate forecast.

Overall, the 3 hr forecast offers greater improvement over the 4 hr forecast in $u$ and $r$ with lesser difference in $h$; this is backed up quantitatively in the spread–error plots (left panel of Fig. 7), where the 3 hr forecast at $T = 40$ hrs shows a smaller
RMSE of the ensemble mean than does the 4 hr forecast in $u$ and $r$.

Next, we analyse the direct impact of assimilating observations at $T = 40$ hrs on this dynamical situation. Fig. 11 shows the 1 hr forecast ensemble initialised at $T = 39$ hrs (left panel) and the analysis ensemble (right panel) at $T = 40$ hrs. Thus, the difference between the ensembles in the left and right panels is due to the assimilation of observations (green circles). First of all, and in reference to the visual interpretation of the 3 and 4 hr forecasts, we observe that the 1 hr forecast also
struggles to capture the intensity of the convection and precipitation around $x = 0.75$, but does certainly improve on the 3 hr forecast (Fig. 10 However, the effect of assimilating observations at this time is evident and promising: the height and rain fields are adjusted, with the analysis ensemble showing good agreement with the verifying nature run, particularly in the region of convection around $x = 0.75$, which also coincides with an observation of $h$. Indeed, the impact of observation proximity on the results is notable, particularly in regions of convection. Good synthetic observations of $h$ that coincide with convective
updrafts have, unsurprisingly, a greater positive impact than those that do not measure the updraft (not shown but evident at other times). Even though the corresponding peak in model rain $r$ does not coincide with an observation of $r$, the ensemble mean of the analysis (cyan line) is greatly improved compared to the ensemble mean of the previous cycle's forecast (red line). This shows the impact of a well-calibrated $\mathbf{P}_e^{\mathrm{f}}$ matrix, in particular the strong positive $h$-$r$ correlations (cf. Figs. 3 and 6), which is able to spread observational information to unobserved parts of the state vector.





The spread of the analysis ensemble is lower than that of the previous cycle's 1 hr forecast, reflecting the increased confidence provided by the assimilated observations. Interestingly, around $x = 0/1$, the threshold heights bring about diverging ensemble members also in the analysis ensemble. The best estimate (i.e., the ensemble mean; cyan line) offers marginal improvement (in $h$ and $r$) on the 1 hr forecast, again illustrating the nonlinear model behaviour and the limitations of the EnKF in dealing with this situation with (highly nonlinear) threshold behaviour. The wind field, on the other hand, shows good improvement

overall. However, we also note the negative impact on the analysis ensemble of the outlying $u$ observations around $x = 0.65$ and $x = 0.85$, highlighting that poor observations can be detrimental to the analysis.

Finally, we note that this subjective verification, by its definition, is open to interpretation and that many other model outputs at other times have also been examined in this process. However, through a critical assessment of Figs. 10 and 11, we have illustrated some of the interesting situations arising in this system, highlighting both the positive and neutral/negative impact

of the data assimilation algorithm in this idealised environment.

## 6   Conclusions

High–resolution 'convection–permitting' NWP models are able to resolve some of the finer–scale features associated with convection and precipitation and are now commonplace in national weather services (cf. Gustafsson et al. (2018)). Idealised models are designed to represent some essential features of a physical system and offer a computationally inexpensive tool for

researching assimilation algorithms. However, their ability to reproduce relevant features of operational NWP systems is not discussed very often and deserves more attention.

The modified rotating shallow water model (modRSW) of Kent et al. (2017) is able to simulate some fundamental dynamical processes associated with convecting and precipitating weather systems, suggesting that it is a suitable candidate for investigating DA algorithms at convective scales. This study exemplifies this further by conducting numerous forecast-assimilation

experiments, providing a critical assessment of their performance, and addressing their relevance for convective-scale NWP.

We have implemented a flavour of the (deterministic) ensemble Kalman filter (Sakov and Oke, 2008), in combination with well-known techniques to tackle undersampling, and have presented a robust assessment protocol with which to establish the suitability of the modRSW model in idealised forecast-assimilation experiments. In particular, we have included RTPS adaptive multiplicative inflation, additive inflation, localisation and self-exclusion, motivated by the Ensemble of Data Assimilations

developed by Bowler et al. (2017), an updated version of which became operational in the Met Office's MOGREPS-G global ensemble in December 2019 (Inverarity et al., 2022).

Experiments have been carried out in the imperfect twin-model setting: the forecast model runs at a coarse resolution, which partially resolves the convection and precipitation fields; synthetic observations are generated by perturbing a higher-resolution nature run, which exhibits "sharper" features in the model fields. We have shown via an extensive tuning process

that there is sufficient error growth in our idealised system, which has an overall observational influence directly comparable to operational systems, for meaningful hourly-cycled DA at kilometre scales. This tuning process involved making *a priori* adjustments to the observing system and iterative adjustments to the filter configuration, while monitoring measures for not





only system performance (ensemble spread, RMSE of the ensemble mean, CRPS) but also relevance (observation influence and error-doubling time). The performance measures have been computed for 3 hr forecasts, to mimic the nowcasting role of convective-scale DA in finding the analysis that produces the best (short-range) forecast.

Our observing system is characterised by the frequency of updates from observations, the total number of observations, imposed observation error and observation spatial density, while basing synthetic observations on the higher-resolution nature run also allows for the inclusion of representation error in the forecast-assimilation system.

The filter configuration is characterised by the size of the ensemble and parameters pertaining to localisation and inflation. Systematically comparing potential set-ups is a crucial process for developing effective forecast-assimilation systems (see, e.g., Poterjoy and Zhang (2015)). In an idealised setting, this process is necessary for assessing both performance and relevance in pursuit of well-tuned experiments. We have synthesised the results of this process in a concise graphical manner that facilitates objective comparison of a large number of experiments, and ascertained how the filter configuration affects the spread–error statistics, CRPS, and overall observational influence. In order to assess a multitude of experimental configurations rapidly, these measures have been computed hourly for 48 hours, and then averaged over time, space, and model variable.

This process identified three candidate experiments (i.e., those with a spread-error ratio close to one and, given this first criterion, minimal RMSE of the ensemble mean and CRPS) to analyse further. Specifically, we focused on the time-dependent spread and RMSE of the ensemble mean of the 3 and 4 hr forecast ensembles for each model variable (see table 3 and Figure 7), and in particular calculated the relative improvement of the 3 hr forecast with respect to the previous cycle's 4 hr forecast for each variable. This confirmed that the spread and RMSE of the ensemble mean are comparable, and therefore that the time-dependent error statistics are adequately captured by the ensemble via its spread. In terms of the improvement in error, the impact of the data assimilation is greatest on the precipitation field, which is a key quantity in operational meteorology (especially nowcasting). The experiment which exhibits the largest improvement (on average) has a relatively low additive inflation factor and an RTPS value $\alpha_{\mathrm{RTPS}} = 0.7$. Our analysis culminated in a critical assessment, both objective and subjective, of this well-tuned experiment chosen from the three short-listed candidates, which further substantiated its relevance.

The headline results are summarised in table 1, which also includes certain aspects of convective-scale NWP and DA that an idealised system should attempt to mimic and ascribes an appraisal of relevance (medium or high) of the idealised system for each aspect. This is by no means a prescriptive or exhaustive list but provides guidance to conducting DA research using idealised models and, in doing so, emphasises the need to assess relevance as well as performance. In our set-up, the forecast model has a grid size of 2.5 km and only partially resolves the convection and precipitation fields, while observations are sampled from a better-resolved nature run. The update frequency (1 hr) is comparable to that of operational convective-scale NWP and error-doubling time statistics ($6-9$ hrs on average) reflect those of convection-permitting models in a cycled forecast–assimilation system, while noting that lower error-doubling times are attainable when using an even higher-resolution nature run. A well-tuned observing system and filter configuration is achieved that adequately estimates the forecast error (with a spread–error ratio close to 1) and has an average observational influence appropriate for convective-scale NWP of about 30%. The filter configuration, with a horizontal localisation lengthscale on the order of 100 km and adaptive multiplicative inflation





(with $\alpha_{\mathrm{RTPS}} = 0.7$), is comparable to filter configurations of operational systems such as those of the Met Office (for its global
ensemble) and Deutscher Wetterdienst.

We recall here our present aim, outlined in the introduction, is to provide convincing evidence that DA experiments using the
modRSW model show consistency with operational forecast-assimilation systems, rather than to compare methods or focus on
a particular aspect of the assimilation algorithm. It should be stressed that there is not a single perfect experiment, but rather
many potential candidate configurations that the user must critically analyse and identify a particular one (or subset) is best
for their specific research goals. It may be that one aspect of the configuration is to be prioritised; for example (and without
loss of generality), in our system we have focused on experiments with a localisation length-scale $L_{\mathrm{loc}} = 1.0$, equivalent to a
Gaspari-Cohn cutoff lengthscale of $2c = 500$ km, and looked at three configurations in more detail. In presenting our results in
this way, we have attempted to lead the reader through the involved process of conducting idealised DA experiments that are
faithful to operational NWP.

By demonstrating a model's suitability for conducting idealised DA experiments and, in particular, showing them to be
relevant for convective-scale NWP, results obtained in an idealised setting have the best chance of scaling up to more-complex
systems. Of course care should be taken when making such claims, and the extra effort made in assessing both performance
and relevance gives greater credence to the outcomes. While there are numerous informative studies using idealised models in
DA research, what is often lacking is a discussion of their relevance to operational NWP. This aspect is a fundamental part of
our study: we have scrutinised our idealised system thoroughly for its relevance and in doing so we have provided confidence
in the use of the model for convective-scale DA research.

*Code and data availability.* The current version of model is available from the project website: https://github.com/modRSW-convective-scale-DA
under a BSD licence. The exact version (v1.0) of the model used to produce the results used in this paper is archived on Zenodo at:
https://zenodo.org/record/7241058#.Y1blvy8Rp-U (Kent et al., 2022b). The data used to produce the plots for all the simulations presented
in this paper are also archived on Zenodo, at: https://zenodo.org/record/7244173#.Y1bl6i8Rp-U (Kent et al., 2022a).

## Appendix A: Equivalence of DEnKF with 'no-perturbation' EnKF with RTPP $\alpha_{\mathrm{RTPP}} = 0.5$

We give here a proof that the 'no-perturbation' EnKF in conjunction with adaptive RTPP inflation using $\alpha_{\mathrm{RTPP}} = 0.5$ (Eq. (12))
is formally equivalent to the Deterministic Ensemble Kalman Filter (DEnKF; Sakov and Oke (2008)). This result is already
being exploited in the Met Office's operational MOGREPS-G Ensemble of Data Assimilations (Inverarity et al., 2022) but, to
our knowledge, has not previously been published prior to Kent et al. (2020). Importantly, this equivalence is preserved when
self-exclusion (section 3.3) is applied to counter the effects of inbreeding. First, note that the analysis step of the EnKF without
perturbed observations ('no-perturbation' EnKF), once self-exclusion is applied to exclude the $j^{\mathrm{th}}$ ensemble member when
calculating the Kalman gain for that member as $\mathbf{K}_{e,j} = \mathbf{P}^{\mathrm{f}}_{e,j}\mathbf{H}^T(\mathbf{H}\mathbf{P}^{\mathrm{f}}_{e,j}\mathbf{H}^T + \mathbf{R})^{-1}$, can be written as:

$$\mathbf{x}^{\mathrm{a}}_j = (\mathbf{I} - \mathbf{K}_{e,j}\mathbf{H})\,\mathbf{x}^{\mathrm{f}}_j + \mathbf{K}_{e,j}\mathbf{y}, \tag{A1}$$





with the ensemble mean $\overline{\mathbf{x}}^{\mathrm{a}}$ being:

$$\overline{\mathbf{x}}^{\mathrm{a}} = (\mathbf{I} - \mathbf{K}_{e,j}\mathbf{H})\,\overline{\mathbf{x}}^{\mathrm{f}} + \mathbf{K}_{e,j}\mathbf{y}, \tag{A2}$$

725  where $\mathbf{I}$ is the $(n \times n)$ dimensional identity matrix. The $j^{\mathrm{th}}$ column of the analysis perturbation matrix $(\mathbf{X}^{\mathrm{a}})_j$ can therefore be expressed as:

$$(\mathbf{X}^{\mathrm{a}})_j = \mathbf{x}_j^{\mathrm{a}} - \overline{\mathbf{x}}^{\mathrm{a}} = (\mathbf{I} - \mathbf{K}_{e,j}\mathbf{H})(\mathbf{X}^{\mathrm{f}})_j, \tag{A3}$$

in which $\mathbf{X}^{\mathrm{f}}$ is the forecast perturbation matrix. The RTPP equation (12) together with the above result yields:

$$(\mathbf{X}^{\mathrm{a}})_j = (1 - \alpha_{\mathrm{RTPP}})(\mathbf{I} - \mathbf{K}_{e,j}\mathbf{H})(\mathbf{X}^{\mathrm{f}})_j + \alpha_{\mathrm{RTPP}}(\mathbf{X}^{\mathrm{f}})_j. \tag{A4}$$

730  For $\alpha_{\mathrm{RTPP}} = \frac{1}{2}$, we obtain:

$$(\mathbf{X}^{\mathrm{a}})_j = (\mathbf{X}^{\mathrm{f}})_j - \frac{1}{2}\mathbf{K}_{e,j}\mathbf{H}(\mathbf{X}^{\mathrm{f}})_j, \tag{A5}$$

which is Eq. (15) in Sakov and Oke (2008), generalized to include self-exclusion.

## Appendix B: Sequential forecast–assimilation algorithm

A compact algorithm for our implementation of one complete cycle (forecast plus analysis) of the DEnKF is summarised here.
735  Throughout, subscript $j$ denotes the $j^{\mathrm{th}}$ ensemble member and subscript $i$ denotes time. Note that prior to the start of the data assimilation algorithm, synthetic observations $\mathbf{y}_j$ are generated by stochastically perturbing the nature run $\mathbf{x}^{\mathrm{nat}}$ valid at the observing time $t_i$:

$$\mathbf{y}_i = \mathbf{H}\mathbf{x}^{\mathrm{nat}} + \boldsymbol{\epsilon}_i^o, \quad \text{where } \boldsymbol{\epsilon}_i^o \sim \mathcal{N}(\mathbf{0}, \mathbf{R}), \tag{B1}$$

$\mathbf{R} = \mathrm{diag}(\sigma_h^2 \mathbf{I}_h, \sigma_u^2 \mathbf{I}_u, \sigma_r^2 \mathbf{I}_r)$, for prescribed error variances $\sigma_{h,u,r}^2$ and identity matrices $\mathbf{I}_{h,u,r}$ with dimension equal to the
740  number of observations of $h$, $u$, and $r$, respectively (so that $\mathbf{R}$ is diagonal with dimension $p \times p$). Unphysical negative synthetic observations of $h$ and $r$ are then reset to zero. A prescribed model-error covariance matrix $\mathbf{Q}$ also has to be estimated.

1. FORECAST STEP:

    i An ensemble of initial conditions $\mathbf{x}_j^{\mathrm{ic}}$ is generated by taking the values from Eq. (15) and adding Gaussian noise for each variable according to $\boldsymbol{\sigma}^{ic}$, as per Eq. (16). Unphysical negative initial conditions for $hr$ are reset to zero if necessary while negative $h$ values (in general very rare and not observed in well-tuned experiments) are reset to
745   0.001.

    ii The model is integrated forward in time. An additive inflation vector $\widetilde{\boldsymbol{\eta}}_j \sim \mathcal{N}(\mathbf{0}, \gamma_{\mathrm{a}}^2 \mathbf{Q})$ is drawn and sample bias is removed by applying Eq. (11). The resulting unbiased model-error vector $\boldsymbol{\eta}_j$ is injected throughout the numerical





integration by dividing it into (small) allocations proportional to the duration of each dynamical time-step $\delta t$. For time-step details, we refer to Kent (2016) and Kent et al. (2017), implemented here with a Courant-Friedrichs-Lewy (CFL) number of 0.5. Therefore, within each forecast step of duration $\Delta t = t_{i+1} - t_i$ at any time $\tilde{t} \in [t_i, t_{i+1}]$, we compute:

$$\tilde{\mathbf{x}}_j^{\mathrm{f}}(\tilde{t} + \delta t) = \mathcal{M}[\tilde{\mathbf{x}}_j^{\mathrm{f}}(\tilde{t})] + \frac{\delta t}{\Delta t}\boldsymbol{\eta}_j, \quad j = 1, ..., N, \tag{B2}$$

with $\tilde{\mathbf{x}}_j^{\mathrm{f}}(t_i) = \tilde{\mathbf{x}}_j^{\mathrm{a}}(t_i)$, where $\tilde{(\cdot)}$ denotes a model state vector concatenating the $n_x$ model variables $h$ followed by $hu$ and then $hr$. In order to ensure that the algorithm does not overshoot the time of the next forecast-assimilation cycle $t_{i+1}$, we take the final time-step to be the reduced value $t_{i+1} - \tilde{t}$ when this is smaller than the optimal $\delta t$ determined by the CFL value. Following the IAU insertion, any negative values of $h$ are reset to 0.001 and negative values of $r$ to zero.

   iii After the one hour assimilation window, the forecasts are continued for another 11 hours, with the additive inflation resampled and injected hourly as in ii, and the timestep again adjusted to produce forecast fields every hour.

2. ANALYSIS STEP:

   i Each $j^{\mathrm{th}}$ $(T+1)$ hr model state obtained from the previous cycle's forecast is transformed into the state vector for assimilation: $\mathbf{x}_j^{\mathrm{f}}(t_i) = \Psi(\tilde{\mathbf{x}}_j^{\mathrm{f}}(t_i))$, i.e., $(h, hu, hr) \mapsto (h, u, r)$.

   ii Compute the innovations $\mathbf{d}_j = \mathbf{y} - \mathbf{H}\mathbf{x}_j^{\mathrm{f}}$ using the forecast states from step 1 and the pre-computed synthetic observations without additional perturbations. Since $\mathbf{H}$ is linear in this setting, the observation location always coincides with the model grid.

   iii Compute the forecast perturbation matrix $\mathbf{X}^{\mathrm{f}}$ and therefore the $N$ forecast-error covariance matrices $\mathbf{P}_{e,j}^{\mathrm{f}}$ (Eq. (7)), each of them computed excluding the $j^{\mathrm{th}}$ ensemble member from step 1, in order to avoid inbreeding.

   iv Apply (model-space) localisation using the Gaspari-Cohn function for a given scaling $L_{\mathrm{loc}}$ to each forecast-error covariance matrix $\mathbf{P}_{e,j}^{\mathrm{f}}$ (section 3.3.1):

$$\mathbf{P}_{e,j}^{\mathrm{f}} \leftarrow \boldsymbol{\rho} \circ \mathbf{P}_{e,j}^{\mathrm{f}}, \tag{B3}$$

compute the $j^{\mathrm{th}}$ Kalman gain matrix $\mathbf{K}_{e,j}^{\mathrm{f}}$ and the subsequent analysis ensemble:

$$\mathbf{x}_j^{\mathrm{a}} = \mathbf{x}_j^{\mathrm{f}} + \mathbf{K}_{e,j}\mathbf{d}_j, \tag{B4}$$

as well as its mean $\overline{\mathbf{x}}^{\mathrm{a}}$.

   v The Deterministic Ensemble Kalman Filter is implemented. The analysis perturbation matrix $\mathbf{X}^{\mathrm{a}}$ is computed with the $N$ members $\mathbf{x}_j^{\mathrm{a}}$ (as per Eq. (7)); using the RTPP implementation of the DEnKF (Appendix A), $\mathbf{X}^{\mathrm{a}}$ is then redefined as:

$$\mathbf{X}^{\mathrm{a}} \leftarrow \frac{1}{2}\mathbf{X}^{\mathrm{a}} + \frac{1}{2}\mathbf{X}^{\mathrm{f}} \tag{B5}$$





vi   Relaxation to Prior Spread (RTPS) is applied using Eq. (13).

vii   The final analysis ensemble is recomputed using the redefined perturbation matrix by applying Eq. (9).

viii   Following ensemble inflation, any negative $h$ values are reset to 0.001 and negative values of $r$ to zero.

ix   Return to step 1: analysis states from step 2viii are transformed back to model variables $h, hu$ and $hr$ with $\tilde{\mathbf{x}}_j^a(t_i) = \Psi^{-1}(\mathbf{x}_j^a(t_i))$ for integration and the sequential cycle continues.

## Appendix C: Gaspari-Cohn function

A common choice for the localising function $\rho$ is the Gaspari-Cohn function, a piecewise rational function (Gaspari and Cohn (1999); their equation 4.10):

$$
\rho(z,c) = \begin{cases} f_1(z/c) & \text{for } 0 \leq z \leq c; \\ f_2(z/c) & \text{for } c \leq z \leq 2c; \\ 0 & \text{for } 2c \leq z; \end{cases} \tag{C1a}
$$

where:

$$
f_1(z/c) = -\frac{1}{4}\left(\frac{z}{c}\right)^5 + \frac{1}{2}\left(\frac{z}{c}\right)^4 + \frac{5}{8}\left(\frac{z}{c}\right)^3 - \frac{5}{3}\left(\frac{z}{c}\right)^2 + 1, \tag{C1b}
$$

$$
f_2(z/c) = \frac{1}{12}\left(\frac{z}{c}\right)^5 - \frac{1}{2}\left(\frac{z}{c}\right)^4 + \frac{5}{8}\left(\frac{z}{c}\right)^3 + \frac{5}{3}\left(\frac{z}{c}\right)^2 - 5\left(\frac{z}{c}\right) + 4 - \frac{2}{3}\left(\frac{z}{c}\right)^{-1}, \tag{C1c}
$$

$z$ is the (absolute) Euclidean distance between two grid points, and $c$ is a length-scale that determines the severity of the localisation. This function (Eq. (C1)) has a similar shape to a half-Gaussian function, noting that $\rho(0) = 1$ and $\rho$ decreases as z increases, with correlations reduced to zero beyond twice the characteristic length-scale.

## Appendix D: Observational Influence Diagnostic (OID)

In the update equation (Eq. (A1)), the gain matrix $\mathbf{K}_{e,j}$ weights the information provided by the observations and prior forecast according to their error covariances. The projection of the analysis estimate into observation space, calculated by premultiplying Eq. (A1) by the observation operator $\mathbf{H}$, is:

$$
\hat{\mathbf{y}}_j = \mathbf{H}\mathbf{x}_j^a = \mathbf{H}\mathbf{K}_{e,j}\mathbf{y} + (\mathbf{I} - \mathbf{H}\mathbf{K}_{e,j})\mathbf{H}\mathbf{x}_j^f. \tag{D1}
$$

The analysis sensitivity with respect to observations, obtained by differentiating Eq. (D1) with respect to $\mathbf{y}$, is given by:

$$
\mathbf{S}_j = \frac{\partial \hat{\mathbf{y}}_j}{\partial \mathbf{y}} = \mathbf{H}\mathbf{K}_{e,j}. \tag{D2}
$$

The OID (Cardinali et al., 2004), generalized for an ensemble system with self-exclusion (section 3.3.3), is defined as:

$$
\text{OID}_j = \frac{\text{Tr}(\mathbf{S}_j)}{p} \tag{D3}
$$

and provides a norm for quantifying the overall influence of observations on each analysis.



*Author contributions.* Tom Kent (TK) has developed and coded the original modRSW model together with its data assimilation algorithm.

Luca Cantarello (LC) has contributed to a number of improvements to both the code and the assimilation scheme. TK and LC have contributed equally to the tuning of the system, the running of the experiments and the analysis of the results. LC has drafted the final version of this manuscript, starting from a previous version written by TK, with all other authors providing feedback on both text and scientific content. Gordon Inverarity (GI) has advised on several technical and scientific aspects related to the data assimilation system and also contributed to some of the coding. Onno Bokhove (OB) and Steve Tobias (ST) have provided scientific advice, assistance, supervision and help throughout

the whole duration of the research. OB set up the research project with partners at the Met Office in 2014.

*Competing interests.* The authors declare that they have no conflict of interest.

*Acknowledgements.* This work stems from a CASE award for TK funded by the Engineering and Physical Sciences Research Council (EPSRC; grant number 1305398) and the Met Office. LC was supported by the NERC SPHERES DTP (NE/L002574/1) award (reference 1925512), co-funded by the Met Office via a CASE partnership. TK acknowledges further support from Prof. Sarah Dance's EPSRC

DT/LWEC Fellowship 'Data Assimilation for the REsilient city' (grant number EP/P002331/1).

The numerical experiments presented in this work were undertaken on ARC4 and ARC3, part of the High Performance Computing facilities at the University of Leeds, UK.





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
