# Peer review of "Experiments with the modified Rotating Shallow Water model (modRSW, v.1.0): assessing the relevance for convective-scale data assimilation research"

_Geoscientific Model Development, 2022_

## Editor Comment (EC1)

**Critical comments**

Since this study is very close to my work, I have read it with great interest and would like to provide some comments for your consideration.

1. I wonder why rotation is omitted in this work. The model can not run with both rotation and orography (it is found in the code that there is no initialization routine for such case)? Since authors attempt to assess the relevance of this study for the real-data convective-scale data assimilation, it would be very interesting and even necessary to show the impacts on rotational and divergent part of dynamics (A similar study but for large-scale data assimilation study has been done by Zeng and Janjic 2016, showing the importance of reconstruction of the rotational part). However, the behavior may be different from the large scale. From this point of view, a study with rotation may be more appealing than with orography (Note: there is no discussion on importance of orography in this study).

2. Experimental settings: 1) The observational resolution is 50 km, which is much coarser than that of observations usually used in convective-scale data assimilation, e.g., resolution of Doppler radars $\sim O(1\ km)$. Therefore, from this point of view, it can not be considered as convective-scale data assimilation. 2) On one hand, to evaluate the skill of spread, a metric called "the spread skill ratio" (Aksoy et al., 2009) is often used, which is calculated as (Spread+Obs. Error)/RMSE. It is optimal if equal to 1. In this study, Spread/RMSE is used, which neglects the observation error and therefore results are probably underestimated. However, on the other hand, considering that the (truncation) model error could be accounted for by the specific additive inflation (the similar method is applied in Zeng et al. 2019, Zeng 2020), additional use of inflation like relaxation methods may result in overestimated spread, which may compensate the deficiency of Spread/RMSE. In my opinion, (Spread+Obs. Error)/RMSE should be used, and the tuning interval of relaxation factor should be efficiently chosen, currently a great part of experiments with larger relaxation factors are not necessary. To my experience, the additive inflation may be already sufficient to maintain the spread (A related study can be found in Zeng et al. 2018).

3. Miscellaneous: 1) There are adaptive RTPP and RTPS but the ones used in this study are obviously not adaptive (see Kotsuki et al. 2017; Ying and Zhang, 2015). 2) How is additive inflation Gaussian? and why are correlations are ignored? We have done similar study (Zeng et al. 2019), those additive perturbations are not Gaussian and we have not explicitly removed the correlations. 3) The typical lead time length for convective-scale data assimilation is 6 hours, instead of 3 hours used in this study. Furthermore, to validate forecasts, it would be more informative to see the plots of variations of RMSE with the lead time.

To sum up, I believe that this framework can be very useful for the community of convective-scale data assimilation, but the presentation needs to be greatly

modified.

**Reference**

Zeng, Y., T. Janjic, A. de Lozar, S. Rasp, U. Blahak, A. Seifert, G. Craig, 2020: Comparison of methods accounting for subgrid-scale model error in convective-scale data assimilation. Monthly Weather Review, 148, 2457-2477.

Zeng, Y., T. Janjic, M. Sommer A. Lozar, U. Blahak, A. Seifert , 2019: Representation of model error in convective scale data assimilation: additive noise based on model truncation error. Journal of Advances in Modeling Earth Systems, 11, 752-770.

Zeng, Y., T. Janjic, A. Lozar, U. Blahak, H. Reich, C. Keil, A. Seifert, 2018: Representation of model error in convective-scale data assimilation: additive noise, relaxation methods and combinations. 2018, Journal of Advances in Modeling Earth Systems, 10, 2889-2911.

Kotsuki, S., Ota, Y., and Miyoshi, T.: Adaptive covariance relaxation methods for ensemble data assimilation: experiments in the real atmosphere, Q. J. Roy. Meteor. Soc., 143, 2001-2015. 2017.

Zeng, Y., T. Janjic., 2016: Study of Conservation Laws with the Local Ensemble Transform Kalman Filter. Quarterly Journal of the Royal Meteorological Society, 142, 2359-2372.

Ying, Y. and Zhang, F.: An adaptive covariance relaxation method for ensemble data assimilation, Q. J. Roy. Meteor. Soc., 141, 2898-2906, 2015.

Aksoy, A., D. C. Dowell, and C. Snyder (2009), A multiscale comparative assess700 ment of the ensemble kalman lter for assimilation of radar observations. Part I: 701 Storm-scale analyses, Mon. Wea. Rev., 137, 1805-1824.

---

## Author Comment (AC1)

**Note to Dr Yuefei Zeng**,

While we were preparing the response to this comment (a first draft of which you will find below), we received an email from Mr. Bastien Chatelon, who made us aware of a significant bug in the data assimilation algorithm published on Github and linked to this paper. All the simulations reported and analysed in the current version of the manuscript are affected by this bug.

Although we believe that the bug does not alter the value of the research conducted, nor the substance and content of our work, we do think that the best way forward at this point is to rerun the tuning of the system from the start and present a new configuration (which will likely entail a new set of optimal parameters and therefore new figures) once the issue has been fixed. This process may take some time, especially as the authors are currently busy in other tasks related to their current occupations and jobs.

As a result, we have decided to withdraw the current version of the manuscript from the peer-review process and to submit a revised version at a later date.

The authors.

**Response to 'comment on gmd-2022-269' by Dr Yuefei Zeng**

We would like to thank Dr Yuefei Zeng for his interesting and relevant comments. We have addressed his remarks below, on a comment-by-comment basis. We have also indicated in the text how we intend to modify or integrate the manuscript accordingly.

**Editor, comment #1:** Since this study is very close to my work, I have read it with great interest and would like to provide some comments for your consideration.

1. I wonder why rotation is omitted in this work. The model can not run with both rotation and orography (it is found in the code that there is no initialization routine for such case)? Since authors attempt to assess the relevance of this study for the real-data convective-scale data assimilation, it would be very interesting and even necessary to show the impacts on rotational and divergent part of dynamics (A similar study but for large-scale data assimilation study has been done by Zeng and Janjic 2016, showing the importance of reconstruction of the rotational part). However, the behavior may be different from the large scale. From this point of view, a study with rotation may be more appealing than with orography (Note: there is no discussion on importance of orography in this study).

**Response to comment #1:** As Dr Zeng has rightly noticed, rotation is not considered in the experiments described in this paper, although it is indeed included in the modRSW model, and has been used both in the case studies considered in the initial model description paper already published (Kent et al., 2017), as well as in the satellite data assimilation experiments conducted by one of the authors in his PhD thesis (Cantarello, 2021, available at this link). In order to consider rotation and orography together, substantial modifications to the numerical scheme currently used would be required. For this reason it was not possible to include

rotation in a straightforward way in the case study illustrated in this manuscript. The possibility of conducting simulations with both rotation and orography would certainly constitute an interesting model upgrade, although we believe it lies outside the scope of this work. We will revise the introduction in the manuscript to clarify more explicitly this limitation and provide a better justification for our choice.

Regarding the orography, we note that it plays a very important role in obtaining continuous generation of  gravity waves which in turn trigger convection and precipitation in our periodic domain without having to reinitialise the experiments. We will clarify the text in section 4.1 of the manuscript to highlight this point better.

**E, c2:** 2. Experimental settings: 1) The observational resolution is 50 km, which is much coarser than that of observations usually used in convective-scale data assimilation, e.g., resolution of Doppler radars ~ O(1 km). Therefore, from this point of view, it can not be considered as convective-scale data assimilation.

**Response to comment #2:** Dr Zeng correctly points out that the common observation density in convective-scale data assimilation systems - when they include radar observations - is higher than 50 km. However, in our experiments we have chosen to simulate only conventional surface observations, for which a 50km station spacing is fairly realistic.
Nonetheless, to address this comment and a similar one made by reviewer 1, we have decided to modify the observing system, including the assimilation of some pseudo Doppler radial wind observations.

**E, c3**: 2) On one hand, to evaluate the skill of spread, a metric called "the spread skill ratio" (Aksoy et al., 2009) is often used, which is calculated as (Spread+Obs. Error)/RMSE. It is optimal if equal to 1. In this study, Spread/RMSE is used, which neglects the observation error and therefore results are probably underestimated. However, on the other hand, considering that the (truncation) model error could be accounted for by the specific additive inflation (the similar method is applied in Zeng et al. 2019, Zeng 2020), additional use of inflation like relaxation methods may result in overestimated spread, which may compensate the deficiency of Spread/RMSE. In my opinion, (Spread+Obs. Error)/RMSE should be used, and the tuning interval of relaxation factor should be efficiently chosen, currently a great part of experiments with larger relaxation factors are not necessary. To my experience, the additive inflation may be already sufficient to maintain the spread (A related study can be found in Zeng et al. 2018).

**Response to comment #3**: We thank Dr Zeng for his remark about the spread skill ratio. Aksoy et al. (2009) measure spread in observation space and compare this with the RMS of innovations based on the ensemble mean in observation space. The latter quantity involves noisy observations and therefore requires the observation error to be taken into account. We agree that observation error needs to be accounted for when observations are included in a performance metric. For example, Bowler et al. (2017) [https://rmets.onlinelibrary.wiley.com/doi/full/10.1002/qj.3004, Appendix B] adjusted the RMSE measure to take account of observation error when verifying against observations. However, in this study, we compare the spread in model space with the RMSE of the ensemble mean, calculated using the nature run as a truth reference. No observations are involved in this metric, so the inclusion of observation error is not required in this instance.
Regarding the use of combined additive and multiplicative inflation, we note that by including both additive inflation and RTPS relaxation in the tuning experiments summarised in Figure

5, we are implicitly showing that optimal performance requires both elements for this experimental configuration. Nonetheless, we plan to add a citation to both Zeng et al. papers (2018, 2019) in section 3.3.2, noting that Zeng et al. (2018) differs in using Q = B (climatological background-error covariances), which combines forecast and observation errors.

**E, c4:** 3. Miscellaneous: 1) There are adaptive RTPP and RTPS but the ones used in this study are obviously not adaptive (see Kotsuki et al. 2017; Ying and Zhang, 2015). 2) How is additive inflation Gaussian? and why are correlations are ignored? We have done similar study (Zeng et al. 2019), those additive perturbations are not Gaussian and we have not explicitly removed the correla- tions. 3) The typical lead time length for convective-scale data assimilation is 6 hours, instead of 3 hours used in this study. Furthermore, to validate forecasts, it would be more informative to see the plots of variations of RMSE with the lead time.

**Response to comment #4:** The aim of this study is to demonstrate that the modRSW model is a credible tool for convective-scale investigations, rather than to produce an optimal data assimilation system. We agree that the additive inflation with a diagonal model-error covariance matrix Q is not optimal, and believe that investigations to find better ways of implementing additive inflation constitute valuable research. Indeed, we hope that the modRSW model and code might provide a framework for such future investigations.

Regarding the definition of nowcasting, we note that the World Meteorological Organization (WMO) Working Group on Nowcasting Research has defined nowcasting as being a period from the present to six hours ahead. Indeed, the document at this link refers to an example of nowcasting in the 0-2 hour range.

To address the final point, we will include a plot of the variation of RMSE with lead time.

**E, c5:** To sum up, I believe that this framework can be very useful for the community of convective-scale data assimilation, but the presentation needs to be greatly modified.

**Response to comment #5:** We thank Dr Zeng for his constructive suggestions.

---

## Author Comment (AC2)

**Note to anonymous referee #1**,

Please note that we have decided to withdraw the current version of the manuscript from the peer-review process and to submit a revised version at a later date. See the response to the editor for more details.

**Response to 'comment on gmd-2022-269' by anonymous referee #1**

We would like to thank the reviewer for their interesting and relevant comments. We have addressed their remarks below, on a comment-by-comment basis. We have also indicated in the text how we intend to modify or integrate the manuscript accordingly.

**Referee #1, comment #1:** This manuscript aims to show the ability of the DA experiments using modRSW to imitate some behaviors of an operational NWP so that one can utilize such inexpensive configurations to study DA in an operational-like environment. The topic is attractive, especially for researchers not in an operational center and who cannot run a real operational NWP due to computational resource limitations. This work is valuable, but the experiments are not well-designed concerning imitating the convective-scale cycling DA. My recommendation is a major revision before publication.

1. The authors perform many experiments and give the experimental configurations relevant to the operational environment in terms of criteria such as grid spacing, ensemble size, localization, inflation, and RMSE. But what are the authors' suggestions on the improvement of localization and covariance inflation? If a new localization scheme or inflation scheme produces good results but cannot satisfy the criteria listed in this manuscript, should they be excluded from the operational NWP development? Showing the ability to imitate an operational-like environment is a good start, but further suggestions are also necessary. At least, the authors should tell readers what criteria should not be violated and what criteria can be adjusted when readers try to improve the DA performance in the simulated operational environment. Is a larger OID (>60%) not acceptable for imitating the convective-scale DA?

**Response to comment #1**: As the referee has correctly stated, the ability to imitate an operational system is an important first step in evaluating the relevance of an idealised model for NWP research. Our paper is primarily interested in demonstrating the credibility of the modRSW model as a tool to investigate convective-scale NWP data assimilation, rather than proposing specific suggestions for the implementation of an Ensemble Kalman filter. Therefore, we don't think that further suggestions are necessary on this matter. For the sake of clarity, we will add some text in the introduction to make this clearer.

Regarding the violation and the adjustment of the tuning and validation criteria mentioned in the paper, the two validation criteria are, for example, somewhat flexible. For OID, the aim is to avoid having a very low (minimal use of observations) or very high (minimal use of the previous forecast) OID. Error doubling times are also somewhat subjective, as they depend on the model resolution and the number and spatial and temporal resolution of observations. In this case, our suggestion is that a doubling time smaller than 12 hours is credible for convective-scale systems. The tuning criteria, on the other hand, should be satisfied by at least some of the experiments. For instance, if no experiment has spread close to the RMSE

of the ensemble mean, then the observing setup should be rejected. We will revisit section 2 of the manuscript to clarify these points further.

**R1, c2**: 2. As far as I know, most operation centers use the variational DA algorithm or the variational-based hybrid algorithm. Only a few centers use the pure EnKF algorithm. Giving a reason for choosing the pure EnKF algorithm to imitate the operational environment is necessary. I do not ask for conducting experiments with a variational DA algorithm, but a brief discussion on the selection should be helpful.

**Response to comment #2**: The referee is right in saying that most operational centres running NWP models use a variational data assimilation algorithm rather than a pure Ensemble Kalman filter. However, the Local Ensemble Transform Kalman filter (LETKF), which is close in concept to an EnKF, is common and popular in the data assimilation community and used operationally by DWD in its KENDA system, for example (see Schraff et al., 2016). In addition, the DA configuration used in our work reflects in some respects the one used at the Met Office for the MOGREPS-G system, as discussed in the recently accepted paper by Inverarity et al. (2023). Besides, the Deterministic Ensemble Kalman filter that we have used in our study is particularly useful because it allows us to implement model-space localisation, which is typical of variational DA systems, together with parallel observation processing.
All in all, we want to stress again that our work is only intended to demonstrate the viability of the modRSW model for data assimilation in general rather than proposing a specific algorithm. We will expand slightly section 3.3 to better clarify these points.

**R1, c3**: 3. With respect to the convective-scale DA, the model and DA configurations are not so representative.
(1) The precipitation procedure in modRSW is more like a cumulus parameterization scheme that estimates the precipitation according to the large-scale thermodynamic environment (see the high correlation between r and h in Figure 6), while a feature of convective-scale NWP is using a microphysics scheme that explicitly simulates the physical procedures in the cloud. The difference in complexity is a gap between a convective-scale NWP and a synoptic-scale NWP. Heavy rain may occur with no large-scale forcing. So I think referring the DA experiments using modRSW as "convective-scale" DA is not proper.

**Response to comment #3**: We agree with the reviewer that the convection and precipitation scheme used in our model is a simple one, and we are aware that it does not include all the features of convective-scale processes. However, adding microphysics would increase the level of complexity of the modRSW model, defeating the purpose of having a simple and inexpensive model. Moreover, the cumulus-convection model developed in Wursch and Craig (2014) – for which the modRSW model constitutes an improvement – was later used in a number of data assimilation studies, all mentioning convective-scale applications (e.g. Ruckstuhl & Janjić, 2018).

**R1, c4**: (2) The observation density is too sparse for the convective-scale DA, especially in the case of assimilating radar and satellite data. The resolution of radar data is often 1 or 2 km. The lack of high-resolution observations is a flaw in the experiments aiming to imitate the convective-scale DA. Using multiscale observations is also a feature of convective scale DA and is not considered or discussed in the manuscript.

**Response to comment #4**: The choice made in this paper was to assimilate only conventional ground observations, for which a spatial density of 50 km is realistic. We are aware that convective-scale data assimilation involves the assimilation of other types of observations, such as satellites and radar reflectivity, however the simplicity of the modRSW model means that some observation types are harder to imitate than others. In this regard, some of the authors have already worked on an isentropic version of the modRSW model, the ismodRSW model, which is better suited for satellite data assimilation experiments. The ismodRSW model (together with its derivation) is discussed in two already published papers (Cantarello et al, 2022; Bokhove et al, 2022) and extensive idealised satellite data assimilation experiments are presented in Cantarello's PhD thesis (2021).
On the reviewer's suggestion, we have modified the observing system in order to imitate the assimilation of Doppler radial wind observations. We will run new experiments in which two bands of u observations are assimilated, alongside the usual h and r observations. We will update the new version of the manuscript with the new results and we will make reference to the above-mentioned papers and PhD thesis regarding the problems of imitating satellite observations using the modRSW model.

**R1, c5**: (3) The precipitation r in the manuscript is more like a simultaneous quantity, e.g., the precipitation rate in a time step. The accumulated precipitation, 3-h or 6-h, is used in real data assimilation. In this respect, the DA configuration in the manuscript does not imitate the real scenarios. This situation should be stated.

**Response to comment #5**: We will state this difference more clearly in the text. We will also note that the rain variable in the modRSW model is defined as a 'mass fraction' and therefore the comparison with accumulated precipitation over time is not straightforward.

**R1, c6**: (4) In convective-scale DA, we must face that many model variables are not directly observed. This issue implies that some model variables must be updated through crossvariable covariance. It is better to show results without r and h observations.

**Response to comment #6**: A set of 'data-denial' experiments, in which each subset of observations was excluded from the assimilation, had already been run in the past, although the results did not end up being included in the first version of the manuscript. Starting from the same DEnKF configuration considered in the paper (same number of ensemble members, same background error and filter parameters), and by excluding one set of observations at a time, our main finding is that assimilating the horizontal velocity is relatively more important to the system than the assimilation of h or r observations. In fact, both the spread and the error grow significantly in a scenario where u is not assimilated, while the overall observation impact decreases slightly. On the other hand, excluding h or r from the assimilation has a less dramatic impact on the performance of the DEnKF, probably because of the strong cross-correlation between these two variables. We will comment on this without providing supporting evidence.

**R1, c7**: (5) The operators of remote observations are often nonlinear, such as those of radar reflectivity and satellite observations. The inaccurate operator is also an issue in the convective-scale DA, but this issue is not discussed.

**Response to comment #7**: The reviewer is correct in saying that observation operators are often nonlinear in operational DA systems. A simpler choice has been made for this paper, also motivated by a stronger focus on the illustrative rather than the representative nature of the DA setup chosen for our experiments. However, as mentioned in the response to comment #4, a nonlinear observation operator for idealised satellite observations has been developed in Cantarello's PhD thesis (2021), in which an upgraded version of the modRSW model (the ismodRSW model) is used.

**R1, c8**: In general, the DA configurations in the manuscript are more suitable for a synoptic-scale DA study; many issues that the convective-scale DA has to face are not discussed. Since it is an idealized study, doing DA experiments with multiscale observations and without r and h observations should not be difficult. Observing the precipitation area with high resolution u observations should result in a much smaller RMSE, similar to radar data assimilation.

**Response to comment #8**: We believe our responses above are enough to justify the use of the modRSW model for 'convective-scale DA' and we hope we were able to address most of the reviewer's concerns. We will modify the manuscript accordingly and thank the reviewer for suggesting the use of high-resolution u observations.

**R1, c9**: L320-323: With respect to imitating LFC with Hc, I have some reservations.

**Response to comment #9**: We acknowledge that this is an approximation of much more complex thermodynamic processes, although this is the same description given in Würsch and Craig (2014).

**R1, c10**: L407: Is $K$ in Equation (19) identical to $K_e$ in Eq. (6b)? If so, use $K_e$ please. If not, what is the difference?

**Response to comment #10**: We thank the reviewer for having noticed this error, which will be fixed in the text.

**R1, c11**: L421: If a new DA method or a new configuration has a much larger influence (OID) in convective-scale DA, what is the authors' suggestion?

**Response to comment #11**: We believe that different OID values would be acceptable as long as they are not too small or too large. Ultimately, idealised models can be used to demonstrate the credibility of a particular approach or configuration, as a precursor of testing in an operational trial with a panoply of diagnostics, rather than just the OID. We will add more text to clarify this point.

**R1, c12**: L606: "the limitations of the EnKF" What are the limitations?

**Response to comment #12**: The main limitations of an EnKF are usually the imposition (or assumption) of Gaussian distributions on an estimated covariance and the adoption of linear or weakly nonlinear error models and observation operators to work properly. We will add this to the text.

**R1, c13**: L611: Should "Fig. 11" be "Figure 11" at the beginning of a sentence?

**Response to comment #13**: We thank the reviewer for noticing this notation issue, which we will also address.

**R1, c14**: L616: It seems that ")." is missed after (Fig. 10.

**Response to comment #14**: We thank the reviewer for noticing the missing parenthesis. This will be fixed in the revised manuscript.

**R1, c15**: L623: "This shows the impact of a well-calibrated $\mathbf{P}^f_e$ matrix" How do authors define a well calibrated $\mathbf{P}$?

**Response to comment #15**: This sentence has not been phrased correctly. What we meant was the impact of 'inter-variable correlations' in the $\mathbf{P}^f_e$ matrix. The text will be modified accordingly.

---

## Author Comment (AC3)

**Note to anonymous referee #2**,

Please note that we have decided to withdraw the current version of the manuscript from the peer-review process and to submit a revised version at a later date. See the response to the editor for more details.

**Response to 'comment on gmd-2022-269' by anonymous referee #2**

We would like to thank the reviewer for their interesting and relevant comments. We have addressed their remarks below, on a comment-by-comment basis. We have also indicated in the text how we intend to modify or integrate the manuscript accordingly.

**Referee #2, comment #1:** The authors aim to show the relevance of using the modRSW model as a tool for mimicking key aspects of convective scale data assimilation in order to justify the transfer of knowledge from a simplified and cheaper setup to an operational configuration. In my opinion, this is a very important and complicated topic which is often overlooked, because it is not straightforward how to tackle it. I think the authors made a good attempt and I encourage the publication of this article, after the authors have considered the following points.

General comments:

1) The topic of this article is very tricky, since much of the relevance of the setup with the modRSW will depend on the purpose of the research. I think the authors should put more focus on the type of research that would be and would not be appropriate with the modRSW. For example, the authors note based on the snapshots that DA can recover the location of convection but struggles with the intensity. We know that operational convective scale DA does have problems with location errors. So for research that seeks to deal with location errors the modRSW may not be suitable. Another important topic among toy model users is non-Gaussianity and positivity constraints on hydrometeors. Is the non-Gaussianity and non-linearity in the modRSW comparable to an operational model? Does the rain get negative values and does it influence the DA results in a similar way as in an operational setup? I encourage the authors to discuss what type of research would be and would not be appropriate with the modRSW.

**Response to comment #1**: The reviewer raises an interesting point regarding the type of research that can be conducted with the modRSW model. A non-exhaustive list of possibilities include: 1) research on strategic choices when assimilating a growing number of satellite observations; 2) research on comparing different data assimilation schemes in the presence of convection and precipitation at the convective-scale; 3) research on the representativity error (for which both the imperfect model scenario and the flexible observing system used in our setup can be particularly useful); and 4) research on data compression techniques (see Fowler, 2019) to optimise the amount of information that can be extrapolated from a growing body of observations.

In this regard, we would like to mention that (1) has been in fact the focus of Cantarello's PhD thesis (2021), in which an updated (i.e. isentropic) version of the modRSW model – the

ismodRSW model – has been used to conduct idealised satellite data assimilation experiments (see also Cantarello et al., 2022, and Bokhove et al., 2022).

**R2, c2**: 2) Comparing this modRSW setup to an operational setup skips the natural step of comparing to an idealised setup with operational model. I think it would be helpful to design a similar idealalised setup with an operational model to compare to the modRSW setup, to distinguish between model-caused differences and any other errors sources that come with the use of real observations. After all, in this work we are interested in the relevance of the modRSW, so we want to isolate its role in the DA experiments. Could the authors provide some thoughts on this matter?

**Response to comment #2**: Unfortunately, the University of Leeds' authors did not have access to an operational model during their research. Also, running the Met Office convective-scale research workflow would have been too complex a task for this research project. Nonetheless, we would like to point out that the data assimilation setup described in our work is largely based on the DA configuration used by the Met Office in their MOGREPS-G system, described in a recently accepted QJRMS paper by Inverarity et al. (2023).

**R2, c3**: L365: Observations → members , right?

**Response to comment #3**: This error will be fixed in the revised manuscript.

**R2, c4**: L370: By discarding negative observations, one creates a positive bias. Is this bias comparable to operational convective scale DA? As mentioned in general comment 1), non-Gaussianity and non-negativity is a popular topic among toy-model users, so I think this point should be explored more elaborately.

**Response to comment #4**: We appreciate the reviewer's comment, and although we have not discarded negative observations, but rather adjusted their values, we agree that this introduces a positive bias to the observation errors.
Regarding the comparison with the bias found in an operational convective-scale DA system, we cannot answer this question quantitatively. However, our approach can be viewed as a simple form of quality assurance analogous to that which is applied to pre-processed observations in operational NWP. Furthermore, a Gaussian error model is not strictly appropriate when applied to non-negative quantities, but is nevertheless convenient in order to apply a Kalman filter.

**R2, c5**: L412: the OID of 0.18, is for real observation experiments I assume. Would we expect a lower value for an idealised setup with operational model? As mentioned in general comment 2) shouldn't that be the value to compare the modRSW setup to?

**Response to comment #5**: The quoted OID was for a global NWP system. The actual value is expected to change depending on the quality of the forecast model (formulation accuracy and resolution, for example) and due to the observation coverage. As such, there is not a single value to compare to, leading to the quoted range, which is an

estimate. We are not aware of a comparable figure for kilometre-scale operational data assimilation systems. We will clarify this point in the text.

**R2, c6**: L421: I don't fully understand how the thresholds of 20 and 40% are chosen, given the numbers mentioned in the previous paragraph.

**Response to comment #6**: We acknowledge that the thresholds for the OID mentioned in the text are somehow arbitrary, although they are intended to lead to a configuration where the prior forecast is the dominant contributor, with observations still having a reasonable influence on the updated forecast. We will clarify this in the text.